# The Role of the Dysregulated JNK Signaling Pathway in the Pathogenesis of Human Diseases and Its Potential Therapeutic Strategies: A Comprehensive Review

**DOI:** 10.3390/biom14020243

**Published:** 2024-02-19

**Authors:** Huaying Yan, Lanfang He, De Lv, Jun Yang, Zhu Yuan

**Affiliations:** 1Department of Ultrasound, Hospital of Chengdu University of Traditional Chinese Medicine, Chengdu 610075, China; yanhuaying1983@163.com (H.Y.); helanfang2006@163.com (L.H.); 2Department of Endocrinology, Hospital of Chengdu University of Traditional Chinese Medicine, Chengdu 610075, China; 3Cancer Center and State Key Laboratory of Biotherapy, Department of Biotherapy, West China Hospital, Sichuan University, Chengdu 610041, China; yangjun4@stu.scu.edu.cn

**Keywords:** JNK signaling pathways, metabolic disorders, neurological diseases, chronic inflammatory diseases, cancer, infectious diseases, JNK inhibitors

## Abstract

JNK is named after c-Jun N-terminal kinase, as it is responsible for phosphorylating c-Jun. As a member of the mitogen-activated protein kinase (MAPK) family, JNK is also known as stress-activated kinase (SAPK) because it can be activated by extracellular stresses including growth factor, UV irradiation, and virus infection. Functionally, JNK regulates various cell behaviors such as cell differentiation, proliferation, survival, and metabolic reprogramming. Dysregulated JNK signaling contributes to several types of human diseases. Although the role of the JNK pathway in a single disease has been summarized in several previous publications, a comprehensive review of its role in multiple kinds of human diseases is missing. In this review, we begin by introducing the landmark discoveries, structures, tissue expression, and activation mechanisms of the JNK pathway. Next, we come to the focus of this work: a comprehensive summary of the role of the deregulated JNK pathway in multiple kinds of diseases. Beyond that, we also discuss the current strategies for targeting the JNK pathway for therapeutic intervention and summarize the application of JNK inhibitors as well as several challenges now faced. We expect that this review can provide a more comprehensive insight into the critical role of the JNK pathway in the pathogenesis of human diseases and hope that it also provides important clues for ameliorating disease conditions.

## 1. Introduction

Cells are able to communicate with each other and adjust their behaviors to adapt to environmental changes. Extracellular signaling is transduced to intracellular organelles through multiple pathways [1]. Among these signaling networks, MAPK cascades are involved in transmitting extracellular signals to intracellular targets [2,3,4]. The MAPK family includes p38, ERK, and JNK. In general, the MAPK signaling pathway mainly consists of MAP3K, MAP2K, and MAPK [5]. Functionally, MAPK signaling regulates several cellular behaviors such as cell proliferation, survival, and death [5]. However, when it is deregulated, MAPK signaling can aggravate the progression of some diseases, such as metabolic disorders, neurological diseases, and various types of cancer [5]. 

JNK (also known as SAPK) is named after c-Jun N-terminal kinase, as it phosphorylates the N-terminal serine residues and activates c-Jun transcription factor [6,7,8]. The JNK family includes JNK1, JNK2, and JNK3. These three isoforms are encoded by the *jnk1*, *jnk2*, and *jnk3* genes, respectively [9]. Like other MAPK family members, JNK also regulates a variety of cell processes such as apoptosis, metabolic reprogramming, and inflammation [9,10,11,12]. JNK activity is strictly regulated in most cells. Therefore, deregulated JNK signaling may cause different, even antagonistic, functions of JNK signaling; for example, impermanent JNK signaling contributes to cell survival, while durative JNK activation leads to cell death [1,13]. 

When it is dysregulated, JNK signaling is harmful to cells and ultimately to the body. JNK over-activation is involved in the development of a variety of diseases. Although the role of the JNK signaling pathway in a single disease such as diabetes has been summarized in several previous publications, a comprehensive review of the JNK signaling pathway and its roles in multiple kinds of diseases is missing. Therefore, in this work, we attempt to comprehensively summarize the recent advances in the JNK research area. To this end, we first discuss the landmark discoveries, structures, tissue expressions of JNK isoforms, and JNK’s activation mechanism. Next, we focus on the effects of the dysregulated JNK pathway on several types of human diseases, including metabolic disorders, neurodegenerative diseases, chronic inflammation and autoimmune diseases, cancers, infectious diseases, and other diseases. Lastly, we summarize the current therapeutic strategies for targeting JNK in treating these diseases. We expect that this review can provide a more comprehensive insight into the critical role of the JNK pathway in multiple kinds of human diseases and hope that it also provides important clues for ameliorating disease conditions.

## 2. Overview of the JNK Signaling Pathway

### 2.1. Landmark Discoveries of the JNK Signaling Pathway

JNKs were initially discovered in the early 1990s [14,15,16,17]. Subsequently, the three mammalian JNK isoforms were cloned in succession [17,18,19], and the functional significance of JNKs, including embryonic development and neurodegeneration, was further explored. Then, researchers have taken great efforts to explore the characteristics of JNKs in the past two decades. These efforts included the structural determination of JNK1, JNK2, and JNK3 [20,21,22], the finding of JNK scaffolds [23,24], and the development of small-molecule inhibitors. Landmark discoveries of the JNK pathway were summarized in Figure 1. Notably, both the structural determination of JNKs in complex with their ligands and the exploration of JNK isoform-specific inhibitors have become two important issues in recent years.

### 2.2. Alternative Splicing, Typical Structure of JNK Isoforms, and Their Expression Localizations

JNK is an evolutionarily conserved family of serine/threonine protein kinases. The sequences of the JNK isoform-coding genes are highly conserved through evolution [25]. Alternative gene splicing is now recognized as a mechanism to produce different splicing forms. Correspondingly, alternative splicing of the *Jnk* gene produces multiple isoforms of the *JNK* gene [19,26]. In general, four spliced forms (α1, α2, β1, and β2) are generated from the *Jnk1* and *Jnk2 genes*, respectively, and eight from the *Jnk3 gene* [27,28,29]. 

As shown in Figure 2A, alternative splicing of JNK1-α1/α2/β1/β2 and JNK2-α1/α2/β1/β2 mainly occurs at alternative exon 6, resulting in long and short N- or C-termini, while the sequence differences of JNK3-α1 and JNK3-α2 are the lengths of the N- or C-termini. The JNK isoforms are typical serine/threonine protein kinases, containing all 11 subdomains as indicated (I–XI). Figure 2B shows the protein sequence of human JNK1, in which the protein kinase activation loop is located between domains VII and VIII. The kinase activation loop contains the conserved amino acid motifs DFG and APE. Within this loop is the threonine (T)-Pro (P)-tyrosine (Y) motif. Structurally, JNK1 contains the N-terminal lobe, the C-terminal lobe, and the activation loop. Catalytic sites are located in the activation loop (Figure 2C). Notably, there is one splicing site between subdomains IX and X of the JNK1 and JNK2 genes, and the resulting splice forms show altered substrate specificity (Figure 2D). The second alternative splicing occurs at the C-terminus of the protein, producing proteins that differ in length by 42 or 43 amino acids (i.e., the typical 46 and 55 kDa proteins). These alternative splicing isoforms cause some functional changes, such as changes in alternative substrate binding, changes in protein interaction, and changes in expression levels [30,31]. Thus, the isoforms of JNK have different substrate specificities and can phosphorylate different non-nuclear substrates or downstream nuclear transcription factors [26,27,32]. As for tissue localization, JNK1/2 expresses in any one cell and in various tissues throughout the body, whereas JNK3 mainly expresses in a few tissues, including the brain, testis, heart, and pancreas [18,27,33,34,35,36]. In terms of cellular localization, although JNK MAPKs are located in both the cytoplasm and nucleus of quiescent cells, activated JNK MAPKs can translocate to the nucleus [31]. Specifically, JNK1 and JNK2 mainly occur in the cytoplasm and nucleus [37,38]. JNK3 can occur in the cytoplasm, nucleus, membrane, and mitochondrion [39]. Due to the differences in cellular and tissue localization, different JNK isoforms may play different roles in the onset and progression of different tissue-related diseases.

### 2.3. Structures of JNK Isoforms

The molecular weights of the three JNK isoforms (JNK1, JNK2, and JNK3) are either 46 kDa or 55 kDa. There is a COOH-terminal extension in some JNK isoforms [40]. The crystal structures of the three JNK isoforms were determined, showing over 90% sequence homology [20,21,22]. Structurally, the JNK protein mainly includes three parts, i.e., the N-terminal lobe, the C-terminal lobe, and a flexible segment connecting the two lobes. The N-lobe contains multiple β strands, while the C-lobe is rich in α helices [27]. Furthermore, the former mainly contains several glutamate–aspartate (ED) domains, while the latter chiefly consists of the common docking (CD) domain [27,41]. The flexible peptide segment provides the kinase activity center containing a Thr-Pro-Tyr motif, which can be phosphorylated by MAP2K kinase. Additionally, JNK kinase includes a kinase activation site and a highly conserved ATP-binding site, which causes a lack of selectivity for small-molecule inhibitors [27]. Figure 2C shows the typical structure of the human JNK1 protein. Recently, the crystal structures of JNK1 bound to the MKK7 docking motif (D2 peptide) [42], JNK1 in complex with ATF2 [43], and AMP-bound human JNK2 [44], as well as JNK3 complexed with thiophene–pyrazolourea derivatives [45], were also reported and shown in Figure 3A, B, C, and D, respectively. These structural determinations will help explore novel JNK inhibitors.

### 2.4. General Activation Mechanism of the JNK Signaling Pathway

JNK belongs to the prodirected serine/threonine kinase family and can be activated by diverse external factors, including cytokines, growth factors, ROS, heat shock, shear stress, pathogens, and drugs [36,46]. Like other MAPK pathways, the JNK pathway is mainly composed of three components, i.e., MAP3Ks (e.g., ASK1), MAP2Ks (i.e., MKK4 and MKK7), and MAPK (JNK1/2/3). As shown in Figure 4, when a cell is stimulated by extracellular stresses, MAP3K (i.e., MEKK1) is first activated in the JNK pathway. Then, the activated MAP3K phosphorylates MKK4 and MKK7, two members of the MAP2K family, which finally phosphorylate and activate the MAPKs (JNK1, JNK2, and JNK3) [28]. Like other MAPK family members, JNK is activated when the Thr-Pro-Tyr tripeptide motif (T-X-Y motif) in the flexible segment is dually phosphorylated by MAP2K. For example, JNK1 is activated when dual phosphorylation occurs on Thr-183 and Tyr-185 [9,26,28]. Although the process of JNK activation is relatively clear, an important question whether the activation of JNKs is due to an increased level of kinase activity or due to an elevated level of JNK proteins need to be addressed. Thus, we decided to discuss this issue in the following several paragraphs.

Previous studies have shown that the catalytic activity of MAPK protein kinase is tightly regulated through the activation segment that is phosphorylated by other kinases to facilitate catalytic activity [47,48]. In general, a kinase will phosphorylate the activation segment of a downstream kinase, allowing the downstream kinase to further propagate a signal. Importantly, phosphorylation of the activation segment at the primary phosphorylation site stabilizes the kinase in a conformation suitable for substrate binding. Kinases may also have secondary phosphorylation sites in their activation segment, which may enhance their activity. Secondary phosphorylation may also aid in the recruitment of substrates by changing the conformation of a kinase to facilitate substrate binding. Once a kinase is stabilized and activated, the catalytic domain identifies a specific substrate and phosphorylates the substrate via its active site [47]. In other words, phosphorylation of the activation segment plays an important role in the regulation of kinase activity. For example, the Ste20-like kinase (SLK), a JNK upstream kinase, can phosphorylate JNK. Luhovy et al. found that, compared with WT-SLK (wild-type SLK), mutations in serine (Ser-189) and threonine (Thr-183) residues in the activation segment of SLK significantly reduce its kinase activity. That is, the T183A, S189A, and T183A/S189A mutants show reduced in vitro kinase activity of SLK. Wild-type SLK, but not mutants, increases activation-specific phosphorylation of JNK kinase [48]. These findings suggest that the phosphorylation of serine (Ser-189) and threonine (Thr-183) residues in the activation segment of SLK determines its kinase activity, which is responsible for phosphorylating its downstream JNK.

As discussed above, JNK belongs to the mitogen-activated protein kinase (MAPK) family. Similar to SLK, JNK activation requires the phosphorylation of both Thr and Tyr residues in its Thr-Pro-Tyr motif in the activation loop between VII and VIII of the kinase domain. For example, JNK1 activation requires the dual phosphorylation of Tyr-185 and Thr-183 in the Thr-Pro-Tyr motif, which is sequentially catalyzed by MKK4 and MKK7, respectively. Interestingly, MKK4 shows a striking preference for the tyrosine residue (Tyr-185), and MKK7 shows a significant preference for the threonine residue (Thr-183) [49,50,51]. From these findings, we can draw the conclusion that, similar to SLK, activation of JNKs also requires the phosphorylation of both Thr and Tyr residues in the activation loop.

Previous studies have shown that the phosphorylation of JNK at Thr-183 and Tyr-185 (p-Thr183/Tyr185) represents its activation form, which exerts the kinase activity of JNK and can phosphorylate its substrates, including c-Jun [52,53,54]. For example, Christopher et al. have recently reported that the phosphorylated JNK (p-Thr183/Tyr185) shows kinase activity on c-Jun and phosphorylates the c-Jun protein at four residues within its transactivation domain (TAD). Among these residues, Ser-63 and Ser-73 are phosphorylated by JNK more rapidly than Thr-91 and Thr-93. Interestingly, different c-Jun phosphorylation states exert different functions: Unphosphorylated c-Jun recruits the MBD3 repressor; Ser-63/73 doubly-phosphorylated c-Jun binds to the TCF4 co-activator, whereas the fully phosphorylated form disfavors TCF4 binding, attenuating JNK signaling. That is, c-Jun phosphorylation encodes multiple functional states that drive a complex signaling response from a single JNK input [54]. 

To the best of our knowledge, no report has shown that the activation of JNKs is directly related to an elevated level of JNK proteins. However, Bildik et al. found that siRNAs targeting JNK, which significantly downregulated JNK expression, led to a significant downregulation of p-c-Jun (Ser-63) and p-c-Jun (Ser-73) [53], implying that the expression level of JNK had an effect on the activation of JNK. Additionally, Yue et al. found that JNK was activated early during stress-induced apoptosis [55,56], and sustained JNK activation accelerated the apoptotic program. Importantly, when caspase-3 was activated during the apoptotic process, JNK was proteolyzed at Asp-385, increasing the release of cytochrome c and caspase-3 activity, thereby creating a positive feedback loop [56]. These studies indicate that the amount of JNK protein may affect its activation. Based on these findings, we think that the activation of JNKs is due to increased kinase activity; however, the expression level of JNK proteins may have an effect on the activation of JNKs.

In the upstream of the JNK pathway, MKK4/7 is regulated by its upstream MAP3Ks. Similarly, MAP3Ks are also activated upon the phosphorylation of specific serine or threonine residues [26]. The common MAP3K members mainly include MLK, ASK1, TAK1, and TPL2 [9]. Notably, scaffold proteins, for instance, JIP1-3, POSH, and IKAP [57,58], facilitate the sequential phosphorylation cascade, while dual specificity protein phosphatase (DUSP) deactivates this pathway through dephosphorylation [28,59]. Additionally, MEKK1 upstream kinases such as NIK, HPK, and GCK are also involved in regulating the activation of the JNK pathway through the phosphorylation of MEKK1. These findings suggest that DUSPs negatively, while HPK and GCK positively, regulate JNK signaling.

The mechanism by which external factors activate JNK is relatively clear. However, how pathogens cause the activation of JNK is not fully understood. Thus, we take some viruses as examples to illustrate which factors of pathogens are decisive for activating the JNK pathway. Previous studies have shown that JNK can be activated by some viruses, such as influenza A virus and HIV-1 [60,61,62]. Mechanistically, most of these viruses use their viral proteins to activate JNK signaling. For instance, the NS1 protein of influenza A virus can activate JNK. Moreover, the amino acid residue phenylalanine (F) at position 103 of NS1 is decisive for JNK activation [60,63]. Similarly, the Tat protein of HIV-1 (human immunodeficiency virus, type 1) activates JNK signaling through a Nox2-dependent manner [62]. SARS-CoV-2 virus also uses its spike protein to activate the JNK pathway through toll-like receptor signaling [64]. Based on these findings, we think that viral proteins are critical for the activation of JNKs. 

Once activated, JNK then phosphorylates its downstream target proteins, which are involved in regulating different cellular responses. Up to now, JNK can phosphorylate almost 100 well-defined substrates [59]. These downstream targets mainly include scaffold proteins (e.g., JIP1), mitochondrial proteins, cytoskeletal proteins, transcription factors (e.g., ATF2), and transmembrane receptors [59,65]. Due to the involvement of these substrates in signaling transduction [59], the activated JNK can regulate a number of cell processes, such as cell differentiation, proliferation, and survival, which play different roles in many types of diseases, such as cancer, metabolic disorders, neurodegenerative diseases, chronic inflammation, autoimmune diseases, and infectious diseases. Notably, several JNK protein complexes are also involved in JNK signaling and some diseases. For instance, the complex composed of ACID, JIP1, and JNK1, the complex composed of JIP1 and JNK1/2/3, and the complex composed of Sab and JNK1/2/3 are also involved in JNK signaling transduction and are critical for neurodegenerative diseases, autoimmune inflammatory diseases/glomerulosclerosis, and non-alcoholic steatohepatitis, respectively. The TAT-JIP peptide and Notch 1-IC (Notch intracellular domain) can be used for ameliorating autoimmune inflammatory diseases and glomerulosclerosis, respectively, through a competitive inhibition of the interaction of JIP1 and JNK1/2/3. 

As for the transcription factors phosphorylated by JNK, it is worth noting that c-Jun is an exclusive JNK MAPK substrate. However, JNK can also phosphorylate and activate other transcription factors. ATF2 is a JNK MAPK substrate that heterodimerizes with c-Jun and stimulates the expression of the *c-Jun* gene [31]. Therefore, through the activation of both c-Jun and ATF2, JNK can regulate the abundance and activity of c-Jun. Elk-1 is another direct JNK MAPK target, whose product forms the AP-1 heterodimer with c-Jun. In the cases of the transcription factors ATF2 and Elk-1, it should also be noted that other protein kinases can lead to their phosphorylation and activation. Specifically, ATF2 can also be phosphorylated by p38 MAPKs, whereas Elk-1 may also be phosphorylated by both ERK and p38 MAPKs [31]. Thus, there is the possibility of cross-talk between the MAPK pathways at the levels of their transcription factor substrates. These findings show that c-Jun may be the common target for JNKs and that JNK MAPKs can phosphorylate various targets to regulate their activity, thus eliciting important biological effects.

In addition, from the activation mechanism of the JNK signaling pathway described above, we can draw the conclusion that the commonalities of JNKs include, but are not limited to, the following elements: (1) different JNK isoforms have a similar structure (e.g., N-lobe, C-lobe, and activation loop); (2) the JNK isoforms all belong to the typical serine/threonine protein kinase, which contains all 11 subdomains; (3) all of the JNK isoforms can phosphorylate c-Jun; (4) they can mediate cell survival or cell death; and (5) JNK isoforms can be negatively regulated by DUSPs or PP2A and regulated by JIP. 

Next, we would like to briefly summarize the regulation of JNK signaling. As we know, JNK belongs to the serine/threonine protein kinase family. As an enzyme, the kinase activity of JNK is strictly regulated in the cell. When the JNK signaling pathway is disturbed, i.e., the kinase activity affected by deviations in the common pathways of activation and inactivation within cells, the deregulated JNK signaling is harmful to cells. The JNK signaling pathway can be disturbed by external factors, such as cytokines, growth factors, ROS, pathogens, and drugs. The abnormal expressions of some genes in the JNK pathway or mutations can also disturb JNK signaling. For instance, the constitutively active mutant of MKK4 can cause sustained activation of JNK, while their dominant-negative mutants can inactivate JNK kinase. By contrast, DUPS overexpression in cells can lead to the inactivation of JNK. In addition, JIP-mimicry peptides can inactivate JNK by interfering with the interaction of JNK and JIP. For example, over-expressed JIP-1 can inhibit JNK MAPK signaling either by inhibiting JNK activity or by altering the subcellular localization of JNK [31]. Interestingly, no report, to the best of our knowledge, has shown that mutations of JNKs themselves lead to the activation of JNKs; however, the constitutively active mutations of some upstream kinases (e.g., TAK1 or MKK4/7) or other molecules (e.g., agouti-related peptide) can activate JNKs (AgRP) [66,67]. In addition, small-molecule inhibitors can inactivate JNK kinase by occupying the kinase catalytic center, inhibiting the kinase activity, or interfering with the JNK–JIP interaction. Due to the complexity of the regulating mechanism of JNK activation and the damage to cells caused by deregulated JNK signaling, deregulated JNK signaling may play a critical role in human diseases. In the following section, we will discuss the role of dysregulated JNK signaling in human diseases.

## 3. The Role of the Dysregulated JNK Pathway in Human Diseases

As discussed above, JNK activity is strictly regulated within cells. Once the JNK pathway is dysregulated, it is harmful to cells and ultimately aggravates disease conditions. Dysregulated JNK signaling contributes to a variety of diseases involving metabolic disorders, chronic inflammation, autoimmune diseases, neurodegeneration, cancer, infectious diseases, and other diseases such as ischemia/reperfusion injury, hearing loss/deafness, and kidney fibrosis. In this section, we mainly discuss the role of the dysregulated JNK signaling pathway in these diseases. 

### 3.1. The Role of the Dysregulated JNK Pathway in Metabolic Disorders

Metabolic syndrome, a metabolic disorder-associated disease, mainly includes obesity, type 2 diabetes (T2D), cardiovascular disorders, and non-alcoholic steatohepatitis (NASH). The dysregulated JNK pathway plays a role in the onset and progression of metabolic syndromes.

#### 3.1.1. The JNK Pathway in Obesity

It is well known that obesity is caused by excessive fat accumulation [68,69]. Obesity often leads to several health problems, such as insulin resistance and T2D [70], fatty liver disease, cardiovascular disease, and cancer [71]. Currently, obesity-driven diabetes is becoming a global metabolic disorder [26]. Although the mechanisms for obesity-triggering T2D are not fully understood, inflammatory kinases play a role in insulin resistance [68]. As an inflammatory kinase, JNK is activated by pro-inflammatory cytokines and mediates the transition from obesity to T2D [68]. 

Previous studies have shown that obesity is a chronic inflammatory disease [36,68]. The evidence is as follows: a large number of macrophages are observed in the visceral adipose tissue. Moreover, macrophages in adipose tissues also secrete pro-inflammatory cytokines, including TNFα and IL-1, which then activate JNK kinases [36]. In addition, JNKs are also stimulated by high-fat diets and are activated by TNFα and IL-1 [26,72,73]. Obesity is also associated with the increase in adipocyte mass, adipokine secretion, and free fatty acids (FFAs), all of which can activate JNK kinases [27]. For instance, FFAs cause JNK activation in cultured 3T3-L1 cells [74]. In adipose tissue, FFAs activate JNK by promoting the activation of the Src–MLK axis [75,76]. Additionally, the scaffold protein JIP1 has been shown to link the Src–MLK-JNK axis with the FFA–JNK signaling axis [75,76]. Importantly, JIP^−/−^ and MLK^−/−^ mice were protected from obesity, showing that the JIP1-mediated Src–MLK signaling axis is needed for obesity [75,77,78,79]. Overall, these findings suggest that the JNK pathway plays a critical role in obesity.

#### 3.1.2. The JNK Pathway in Insulin Resistance and T2D

High-fat diets (HFDs) trigger JNK over-activation, which leads to insulin resistance [80]. The opinion that JNK functions in insulin resistance is supported by the following observations: (1) JNK over-activation is observed during obesity, and mice with a JNK1 deficiency display a significantly improved insulin sensitivity level [72,80]; (2) JNK1 knockout mice are protected from obesity [80,81]; (3) mice lacking JNK2 also display an improved insulin sensitive phenotype [81]; and (4) there are some mutations in the JIP1 coding gene (JNK’s scaffold) in patients with type II diabetes [82]. 

As for the action mechanism, Solinas et al. provided four distinct mechanisms to explain the involvement of JNK in T2D: (1) the phosphorylation of IRS1/2; (2) contribution to metabolic inflammation; (3) negative regulation of the TSH–thyroid hormone axis; and (4) inhibition of PPARa-FGF21 signaling [80]. These four mechanisms are summarized in Figure 5. The first mechanism is supported by the report that anti-TNFα treatment ameliorates insulin sensitivity [83] and also by the findings that TNFα-activated JNK directly phosphorylates IRS1/2, causing insulin resistance [80,84]. Specifically, Tanti et al. found that phosphorylation of IRS1 by JNK inhibits its interaction with insulin receptor (IR), causes defective IRS1 tyrosine phosphorylation, and importantly, abrogates IRS downstream PI3K-AKT signaling, thus promoting FOXO1-mediated gluconeogenesis, and finally causing insulin resistance (Figure 5A) [85,86,87]. Notably, both JNK1 and JNK2 can phosphorylate IRS1 [81], and JNK is also able to phosphorylate IRS2 [86,88,89]. 

Han et al. found that JNK activity in macrophages of adipose tissue contributes to obesity-induced insulin resistance and inflammation [90]. Mechanistically, the activated JNK1/2 promotes c-Jun, ATF-2, and ELK1 to bind to the promoters of pro-inflammatory cytokine-coding genes such as *IL-6*, subsequently causing the transcription of these genes, finally resulting in the secretion of pro-inflammatory cytokines from macrophages. Similarly, Perry et al. found that JNK signaling promotes the secretion of IL-6, causing an increase in FFAs that further facilitate the liver’s ability to synthesize glucose, finally resulting in insulin resistance [30]. Together, these studies indicate that JNK activation exacerbates the progression of T2D through the promotion of metabolic inflammation (Figure 5B). In regard to the regulation of the TSH–thyroid hormone axis by JNK, Belgardt et al. discovered that the expression levels of T3 and TSHβ in the pituitary were increased in HFD-feeding mice, whose JNK1 is deficient in the central nervous system, indicating that JNK inhibits the TSH–thyroid hormone axis. Moreover, body mass and liver triglyceride content were also reduced in these mice [91]. Similarly, Sabio et al. found that JNK1 deficiency in the CNS also caused a decrease in adiposity in HFD-feeding mice. Taken together, these studies showed that HFD-stimulated JNK1 promotes adiposity by negatively regulating the TSH–thyroid axis [92]. Moreover, the activated JNK promotes AP-1 transcription factors, which include c-Jun and c-Fos, to promote Dio2 gene transcription and expression. Then, Dio2 makes T4 convert into T3, which further inhibits TSHβ gene expression (Figure 5C) [93]. Subsequently, the reduced TSHβ results in a reduction of thyroid hormone and an increase in adiposity, which finally leads to insulin resistance. Notably, Vernia et al. found that HFD-feeding-activated JNK3 in hypothalamic neurons is needed for controlling appetite for food [94]. Interestingly, in contrast to control animals, HFD-feeding JNK3 knockout mice are apt to develop obesity and subsequent insulin resistance, suggesting neuron-specific JNK3 plays a role in improving insulin resistance [94].

Vernia et al. found that hepatocyte-specific JNK1/2 knock-out improved insulin sensitivity and significantly ameliorated fatty liver in a FGF21-dependent manner [95,96]. Mechanistically, JNK first stimulates c-JUN to bind to the promoter of the NcoR1 gene and promotes the transcription and subsequent expression of NCOR1. As a transcriptional corepressor, NCOR1 further represses the PPARα and RXR-mediated PPARα response genes, such as FGF21 expression, thus affecting fatty acid oxidation and ketogenesis, which finally result in insulin resistance (Figure 5D). In addition to these above mechanisms, JNK activity can affect insulin secretion from pancreatic β-cells. In the pancreatic β-cell, the elevated FFAs in plasma during obesity will result in the sustained activation of JNK, which, in turn, negatively regulates insulin secretion and finally causes β-cell dysfunction and death [68,97]. Collectively, these studies demonstrate that JNK plays an important role in obesity and type II diabetes. 

#### 3.1.3. JNK in Atherosclerosis

Atherosclerosis is a multi-step systemic inflammation and metabolic disease, which ultimately leads to myocardial infarction or stroke [36]. Both obesity and diabetes are known to contribute to atherosclerosis. Atherosclerosis onset and progression involve the following events: (1) An initiating stimulus, including vascular injury, hypercholesteremia, or chronic inflammation. These stimuli cause endothelial cell dysfunction and/or apoptosis, thus increasing vessel wall permeability to lipids and local inflammation. Under these conditions, endothelial cells then express cytokines that recruit monocytes and leukocytes to the area; (2) monocytes then trans-migrate across the blood vessel wall and differentiate into macrophages. As the vessel is lipid-laden, the macrophages ingest the low-density lipoproteins (LDLs), and then turn into foam cells. These foam cells further promote lesion progression by increasing local inflammation and ROS production; (3) under the actions of endothelial cell-secreted cytokines, T and B lymphocytes are recruited to the plaque. Smooth muscle cells are also recruited to the lumen and begin to proliferate and secrete collagen, elastin, and other extracellular matrix proteins [36]. Although multiple cell types contribute to this complex process, elevated JNK activity is observed in each of these steps [36]. As mentioned above, JNK1 or JNK2 plays an important role in inflammation and cytokine production. Moreover, JNK1 or JNK2 is expressed in all of the cell types relevant to the onset and progression of atherosclerosis: endothelial cells, smooth muscle cells, macrophages, and T cells, all of which are associated with the formation and development of atherosclerosis. Specifically, JNK1 promotes apoptosis in the endothelium after chronic inflammation and promotes atherosclerosis. Similar to the endothelium, JNK1 in bone marrow-derived immune cells (including monocytes) also promotes apoptosis after chronic inflammation, which leads to less atherosclerosis in mice. JNK2 knockout mice show less atherosclerosis through a reduced number of foam cell formation. These studies suggest that JNK is involved in inflammation and cytokine production, mediates the apoptosis of the endothelium, immune cells, and foam cells, and has different functions in different cells (macrophages, T cells, and endothelial cells). Previous studies have shown that JNK over-activation promotes the progression of atherosclerosis [36,98]. Furthermore, Ricci et al. found that JNK2 ablation can mitigate atherosclerosis progression, implying that JNK2 can promote atherosclerotic lesion formation [99]. However, Babaev et al. demonstrated that a hematopoietic cell-specific JNK1 deficiency promoted atherosclerosis in LDLR^−/−^ mice. By contrast, Nofer et al. found that JNK promoted atherosclerosis onset. Similarly, Amini et al. found that JNK1 ablation protected atherosclerosis formation [100]. These studies suggest that JNK is critical for the onset and progression of atherosclerosis.

#### 3.1.4. JNK in Non-alcoholic Fatty Liver Disease (NAFLD)

NAFLD includes non-alcoholic fatty liver (NAFL) and non-alcoholic steatohepatitis (NASH) [101]. Hypertension and cardiovascular diseases, hepatic gluconeogenesis-triggered hyperglycemia, and insulin resistance-caused hyperinsulinemia all lead to NAFL disease [101]. Although the mechanism is not fully understood, JNK activation in response to hepatic metabolic stresses, including inflammation, cytokines, de novo lipogenesis, and lipolysis, has been identified as a reason for NAFL/NASH [102,103,104,105,106]. 

It is well known that the liver contains about 60% parenchymal cells and over 30% non-parenchymal cells, including hepatic stellate cells. Among these cells, hepatocytes, a type of parenchymal cell, are the cornerstone of the liver. Hepatocytes express three JNK isoforms [107]. Diets, especially hyper-nutrition-type diets, including HFD, easily activate hepatic JNK [96]. Mechanically, HFD first activates JNK in the hepatocytes [108,109]. Then, the activated p-JNK further phosphorylates SAB, a mitochondrial outer membrane protein, resulting in mitochondrial ROS production, which further activates the ASK1-MKK4/7-JNK axis [107,110], thus forming a JNK-SAB-ROS-ASK1-JNK feedback loop, finally resulting in sustained JNK activation. This feedback loop is considered a critical mechanism in the development of hepatic steatosis [30,51]. The important role of JNK in NASH is proved by the following reports: ASK1 knockout mice are protected from hepatic steatosis [101,111]; MLK knockout mice also reduce the level of triglycerides [112]; similarly, JNK knockout mice avoid HFD-triggered NASH [113]. 

### 3.2. The Role of JNK Signaling in Neurological Diseases

JNK signaling exerts different roles in neurogenesis, axonal growth, axonal transport, and brain metabolism [114]. The deregulated JNK pathway leads to developmental axonal defects. Moreover, the over-activation of the JNK pathway, which means the kinase activity of JNK is constitutively increased, also contributes to CNS pathologies and motor neuron (MN) diseases [8]. Here, we briefly summarize the function of JNK in CNS pathologies such as glioblastoma progression, neurodegenerative diseases, and CNS regeneration/repair after an injury. 

#### 3.2.1. The JNK Pathway in Glioblastoma Progression

The JNK pathway is highly active in the central nervous system (CNS) as compared with other tissues. Previous studies revealed a dual role of JNK in cell death and survival, which is very important for glioblastoma (GB) tumorigenesis and neurodegeneration [115,116,117]. GB is a malignant brain tumor [114,118]. Over-activation of the JNK pathway is a hallmark related to glial cell proliferation and cancer stem cell-like properties [117,119,120]. Recent findings showed that JNK activity correlates with GB aggressiveness and that JNK promotes GB progression and infiltration [117,121]. Mechanically, JNK activation leads to upregulation of matrix metalloproteases (MMPs) [117], causing extracellular matrix degradation and promoting GB cell infiltration [117]. Interestingly, the progressive increase of JNK pathway activation in GB samples is associated with the localization of the Grnd receptor’s ligand in glial cells [121]. Recently, Jarabo et al. found that GB cells can secrete Impl2 to induce neuronal changes, contributing to tumor progression. Importantly, this process is also regulated by JNK activity [122]. Together, these studies indicate that GB cells use different ways to activate the JNK signaling pathway, further promoting GB progression. 

#### 3.2.2. The JNK Pathway in Neurodegenerative Diseases

Parkinson’s disease (PD), Alzheimer’s disease (AD), and Huntington’s disease (HD) are the three most common neurodegenerative diseases [26]. CNS-specific JNK3 has been shown to be a mediator for these neurodegenerative diseases. In AD patients, there are many diffuse plaques in cortical and hippocampal neurons [26,123]. These plaques mainly contain β-amyloid peptide (Aβ) and hyperphosphorylated Tau protein. Aβ induces neuronal apoptosis through JNK-dependent downregulation of Bcl-w, resulting in cortical neuron apoptosis [124]. JNK is also responsible for phosphorylating Tau to promote tangle formation [125]. In addition, JNK3 over-activation promotes AD progression as JNK3-deficient mice display synapse regrowth and a decrease in Aβ [114,126]. These mice are protected from Aβ-induced neuronal cell apoptosis through a JNK3-mediated AP-1-dependent FasL manner [123,124,127]. 

JNK also mediates PD progression. Choi et al. found that neurotoxin-induced JNK3 activation mediated dopaminergic neuron death in the substantia nigra [128]. JNK over-activation induces apoptosis of dopaminergic neurons [129]. Consistently, JNK3^−/−^ and JNK2^−/−^ mice prevented dopaminergic cell death induced by the neurotoxin MPTP [26]. Mechanistically, JNK-mediated autophagy and apoptosis play key roles in PD progression [129]. Bcl-2 has been identified as a critical protein with the ability to suppress autophagy and apoptosis through inhibiting Beclin-1 and Bax, respectively. Moreover, both JNK and P38 mediate dopaminergic neuron death in PD via apoptosis by increasing the Bax/Bcl-2 ratio. With regard to autophagy, JNK-mediated BCL-2 phosphorylation also suppresses the functions of Bcl-2 in autophagy. Additionally, recent findings indicate that receptor-interacting protein kinase 1 (RIPK1) is upregulated in PD *in* in vitro and in vivo models. RIPK1 promotes cell apoptosis and reactive oxygen species (ROS) production through the activation of the JNK pathway, thus leading to dopaminergic cell death. These discoveries suggest that JNK, together with other factors such as p38, BCL-2, Bax, Beclin-1, and RIPK1, mediate the progression of PD.

HD disease is caused by the degeneration of projection neurons in the striatum and the cerebral cortex. In HD patients, there are some amplifications of glutamine repeat (poly Q) in huntingtin protein (HTT), which trigger protein aggregation and neuronal degeneration [130]. During this process, polyQ-containing proteins activate JNKs. Furthermore, Morfini et al. demonstrated that aberrant HTT would impair axonal transport (FAT) through the JNK-mediated phosphorylation of the kinesin-1 protein [131]. Furthermore, JNK kinase inhibitors conferred neuroprotection in an HD mouse model [132], providing further evidence that JNK functions in HD progression. Taken together, these observations have demonstrated that the JNK signaling pathway is critical for the progress of neurodegenerative diseases. 

#### 3.2.3. JNK in Excitatory Toxicity of Hippocampal Neurons

The expression of JNK3 in the fore- and hind-brain regions is high. Normally, mice treated with kainic acid, a neurotoxic reagent, display significant apoptosis of hippocampal neurons [26]. JNK3 knockout mice also avoid glutamate-induced excitatory toxicity. Similarly, mice knocking in a mutant c-Jun (S63A and S73A) that cannot be phosphorylated by JNK display resistance against glutamate-induced neuronal toxicity [133]. These studies indicate that JNK3 activity promotes neuronal excitatory toxicity.

#### 3.2.4. Abnormal Activation of the JNK Pathway in Multiple Sclerosis

Multiple sclerosis (MS) is a chronic demyelinating neurodegenerative disease of the CNS, manifesting as myelin sheath damage. Zhang et al. found that JNK was overactivated in chronic active MS plaques [134]. Bagnoud et al. also found that the phosphorylated JNK is increased during demyelination-triggered MS [135]. During the progression of MS, the phosphorylation of JNK by ASK1 plays critical roles in multiple processes, such as oligodendrocyte destruction, demyelination, neuroinflammation, immune dysregulation through T cell activation (T cell apoptosis), and oxidative damage. For instance, the TLR-ASK1-JNK pathway is active in glial cells and important for autoimmune demyelinating disorders [136]. In addition to the TLR (Toll-like receptor), other factors, including reactive oxygen species (ROS), oxidative stress, and inflammation, can activate ASK1 during MS progression [137]. These reports indicate that the over-activated ASK1-MKK4/7-JNK signaling axis plays a critical role in the pathogenesis of MS; that is, deregulated JNK signaling contributes to MS progression [134,135,137]. 

#### 3.2.5. JNKs in CNS Damage Repair

Some factors, such as brain stroke and spinal cord injury, often trigger CNS damage. Because the self-repair or regenerative ability of this tissue is limited, the damaged CNS often displays permanent disabilities [114]. Previous studies have shown that JNK signaling has an important function in degeneration and the repair of nerve injuries. That is, retrograde transport of JNK in the injured axons changes the transcription of ATF3, thus affecting axon growth [138,139]. In addition, suppression of JNK signaling delays axonal degeneration [140]. DLK-JNK signaling also contributes to axonal regeneration after spinal cord injury [141,142,143,144]. Additionally, JNK3 interacts with KLF9 to inhibit axon regeneration upon nerve injury [145]. Overall, these reports clearly show the relevance of JNK to regeneration/repair after CNS injury.

### 3.3. JNK Pathway in Chronic Inflammatory Disease

Autoimmune disorders are a group of chronic inflammatory diseases, such as autoimmune arthritis (AA), osteoarthritis (OA), and inflammatory bowel disease (IBD). Here, we take AA, OA, and IBD as examples to demonstrate the function of JNK in chronic inflammatory disease. 

#### 3.3.1. The JNK Pathway in Autoimmune Arthritis

Rheumatoid arthritis (RA), psoriatic arthritis (PsA), and ankylosing spondylitis (AS) are three common autoimmune arthritis (AA) diseases [146]. The common characteristics of these diseases is the destruction of the joints and bones. Interestingly, extensive lymphocyte infiltration occurs in the inflamed joints in AA [146]. Also, pro-inflammatory cytokines, for instance, tumor necrosis factor-alpha (TNF-α), are significantly elevated in serum and joint fluid [146,147]. 

Fukushima et al. found that the phosphorylation of JNK was increased in the joint tissues of mice with inflammatory arthritis [148,149]. Similarly, JNK activation was also observed in synovial tissues from patients with psoriatic arthropathy and RA patients [150,151]. Recently, Li et al. found that JNK signaling was also involved in the pathogenesis of ankylosing spondylitis [152]. Mechanically, JNK is activated by pro-inflammatory cytokines, including TNF-α, causing autoimmune arthritis [146]. Additionally, JNK also regulates the expression of metalloproteases, promoting joint destruction [26], as JNK1/JNK2-deficient rheumatoid arthritis mice were protected from joint damage and reduced the expression level of metalloproteinase [26]. These studies clearly suggested that deregulated JNK signaling plays a role in autoimmune arthritis. 

#### 3.3.2. The JNK Pathway in Osteoarthritis

Osteoarthritis (OA) is a joint degeneration-related disease. The characteristics of OA mainly include several pathologic changes, such as joint inflammation, erosion of articular cartilage, and osteophyte formation [153]. JNK has been shown to aggravate this pathological process. For example, upon activation, JNK promotes cartilage destruction. Specifically, JNK is first activated by the pro-inflammatory cytokine TNFα; then, the activated JNK phosphorylates c-Jun, promoting MMP-13 (a cartilage-degrading enzyme) transcription and expression, which finally causes a decrease in proteoglycan synthesis. Additionally, excessive MMP-13 expression in chondrocytes contributes to cartilage degeneration [26,154]. Additionally, Yang et al. found that the elevated levels of CXCL8 and CXCL11 in the synovial fluids also aggravate OA progression by activating the JNK pathway, which mediates the apoptosis of chondrocytes [155]. 

#### 3.3.3. JNK Functions in Chronic Inflammatory Bowel Disease

Crohn’s disease and ulcerative colitis are two chronic inflammatory bowel diseases (IBDs). Several pro-inflammatory cytokines in the intestinal microenvironments, for example, TNFα, can activate JNK signaling, which accelerates the development of IBD [156]. Previous studies showed that JNK was highly activated in the intestinal tissues from IBD patients [156,157]. Similarly, Mistsuyama et al. found that the active form of JNK was expressed in the nuclei of intestinal cells, macrophages, and lymphocytes, further providing indirect evidence that pro-inflammatory cytokines activate the JNK pathway in IBD [157]. Chromik et al. investigated the function of JNK isoforms in IBD and found that JNK2 deficiency aggravated DSS-induced colitis in mice; however, JNK1 deficiency did not completely block colitis development [158]. By contrast, the JNK inhibitor SP600125 partially protected against colonic injury [156]. 

### 3.4. Pro- and Anti-Oncogenic Roles of JNK in Cancer

The JNK pathway has a dual role in different types of cancers, including hepatocellular carcinoma (HCC), pancreatic cancer, prostate cancer, multiple myeloma and oral cancer, non-small-cell lung cancer (NSCLC), glioblastoma, papilloma, intestinal tumors, breast cancer, squamous cell carcinoma, Burkitt’s lymphoma, and ovarian cancer [1,9,26,57,114,159]. The dual effects of JNK in oncogenesis are summarized in Figure 6. On the one hand, JNK can promote cell death, eliminating pre-tumorigenic cells. On the other hand, it can also stimulate tumorigenesis [26,115]. 

#### 3.4.1. The Dual Role of JNK in Hepatocellular Carcinoma (HCC), Multiple Myeloma, Prostate Cancer, and Oral Cancer

JNK has a tumor promoting or suppressing function in HCC, pancreatic cancer, multiple myeloma, and oral cancer (Figure 6A). JNK1 deficiency in hepatocytes and JNK2 deficiency in the liver promote HCC progress, indicating its role in inhibiting tumor development [159]. Similarly, JNK1 deficiency significantly decreases susceptibility to diethylnitrosamine-induced hepatocarcinogenesis [26]. However, JNK has an oncogenesis role in non-parenchymal cells of the liver through the production of pro-tumorigenic cytokines [159]. Hideshima et al. found that the JNK inhibitor SP600125 arrests MM cells in the cell cycle and causes cell growth inhibition through activation of the NF-κB pathway [160]. However, Sharkey et al. indicated that SP600125 abolished the apoptosis of multiple myeloma cells [161]. These two studies clearly showed the controversial role of JNK in multiple myeloma through different mechanisms affecting cell apoptosis and survival. In the same way, the JNK pathway plays dual functions in prostate cancer [9]. JNK has been shown to contribute to the apoptosis of prostate cancer through endoplasmic reticulum stress, the death receptor-dependent apoptotic pathway, and the mitochondria pathway [9]. By contrast, several studies showed that the JNK pathway is involved in prostate cancer progression [9,162,163]. For example, Sung et al. found that Jazf1 promoted prostate cancer progression by activating JNK signaling in DU145 prostate cancer cells [162]. Similarly, Du et al. discovered that CC chemokine receptor 7 (CCR7) activated the Notch1 pathway, resulting in a significant increase of phosphorylated JNK, which further promoted the migration of prostate cancer cells [163]. In regard to oral cancer, the JNK pathway has been shown to play an oncogenic or tumor-suppressive role through action alone or synergistically with other MAPKs [164]. For instance, Noutomi et al. found that after activation, JNK further enhanced TRAIL-induced apoptosis of oral cancer cells [165]. Kim et al. discovered that JNK, together with ERK, were implicated in ROS-induced apoptosis of OSCC cells [166]. These studies showed that JNK has a tumor-inhibiting function in oral cancer. On the contrary, Gross et al. found that inhibition of JNK suppresses tumor growth in oral cancer by downregulating IL-8 signaling, VEGF activity, and EGFR activation [167], suggesting JNK’s pro-oncogenic role in oral cancer.

#### 3.4.2. The Pro-Oncogenic Role of JNK in NSCLC and Glioblastoma

Emerging evidence suggests that the sustained activation of JNK contributes to the progression of NSCLC and glioblastoma [159]. Since the pro-oncogenic role of JNKs in glioblastoma has been summarized in this review (in the section on JNK’s role in neurological diseases), here, we briefly discuss the role of JNK in NSCLC. Luo et al. found that LINC00958 promoted NSCLC cell proliferation by activating the JNK pathway [168]. Recently, Lin et al. reported that KIAA1429 promoted NSCLC tumorigenesis through the activation of the JNK pathway [169]. These studies showed a pro-oncogenic role of the JNK pathway in NSCLC (Figure 6B). 

#### 3.4.3. JNK Functions as a Tumor Suppressor in Intestinal Tumors, Papilloma, and Breast Cancer

The tumor-suppressive role of the JNK pathway in intestinal cancer, papilloma, and breast cancer is shown in Figure 6C. First, its role in breast cancer was supported by the findings of Cellurale and others that JNK1/2 ablation significantly increases tumor formation in breast cancer [170]. Subsequently, Shen et al. reported that cambogin induced the apoptosis of breast cancer cells by activating JNK signaling [171]. More recently, Itah et al. also found that JNK inhibition contributed to the progression of breast cancer [172]. Similarly, JNK also acts as a tumor suppressor in intestinal cancer and papilloma. For example, Tong et al. found that JNK1 knockout mice spontaneously developed intestinal cancer [173]. Recently, Kwak also reported that isolinderalactone exerted its anticancer effect on oxaliplatin-resistant colorectal cancer cells by inducing JNK-mediated apoptosis [174]. Choi et al. found that JNK1 can phosphorylate Myt1 to induce caspase-3-mediated apoptosis, thus preventing the formation of skin cancer, and that JNK1 knockout mice developed more UV-induced papilloma than JNK wild-type mice [175].

#### 3.4.4. JNK Inactivation Suppresses Tumorigenesis in Ovarian Cancer, Skin Cancer, and Lymphoma

Emerging evidence showed that inhibition of JNK signaling prevented the oncogenesis of skin cancer, lymphoma, and ovarian cancer (Figure 6D) [159]. For instance, JNK2-KO mice exerted suppression of skin tumorigenesis [176], indicating the tumor-promoting role of JNK2 in skin cancer [57,176]. Additionally, Ding et al. found that targeting the JNK pathway suppressed Burkitt’s lymphoma cell proliferation [177]. Yang et al. found that JNK3 inactivation enhanced BH3 mimetic S1-induced apoptosis in cisplatin-resistant human ovarian cancer cells [178]. Similarly, JNK1 inhibition also suppressed ovarian cancer growth [179]. These results suggested the suppressive roles of different JNK isoforms in skin cancer, lymphoma, and ovarian cancer. 

### 3.5. The JNK Pathway in Infectious Diseases

The JNK pathway has been shown to function in multiple infectious diseases, which are caused by pathogens, including viruses, bacteria, fungi, and parasites [61]. Thus, we briefly discuss JNK’s role in these infectious diseases. 

#### 3.5.1. JNK in Viral Diseases

The JNK pathway can be activated by viruses, which, in turn, regulates many viral infectious diseases [180]. The majority of viruses use their encoding proteins to activate JNK signaling. For instance, the NS1 protein of the influenza A virus can activate JNK [60]. Similarly, the Tat protein of HIV-1 activates JNK signaling through a Nox-mediated mechanism [62]. The activated JNK, generally speaking, further supports viral infection and replication [61]. Previous studies have shown that JNK inactivation results in significantly reduced replication of viruses, such as dengue virus [181], influenza virus [182], and veterinary viruses [180]. Mechanistically, JNK promotes apoptosis of infected host cells, accelerating viral infection [61]. In addition, JNK is also important for viral replication via autophagy. In addition to the contribution of JNK to the replication of most viruses, JNKs also negatively regulate the replication of the oncolytic vaccinia virus [61]. 

#### 3.5.2. JNK in Bacterial Infections

Previous studies have shown that JNK can be activated upon bacterial infection in multiple mammalian cell types [40,183]. Correspondingly, bacteria have evolved several mechanisms for activating JNK. For example, lipopolysaccharide (LPS), an outer membrane protein of Gram-negative bacteria, can activate the JNK signaling pathway [184]. Nguyen et al. showed that pneumolysin, a virulence factor of *Streptococcus pneumoniae*, can activate the JNK pathways to induce ATF3 expression [185]. Similarly, *Escherichia coli* (*E. coli*)-produced shiga toxin 1 is able to activate JNK, and subsequently cause the apoptosis of intestinal epithelial cells [186]. In addition, some bacteria themselves can regulate host JNK activity through posttranslational modification of JNK proteins [61]. For instance, AvrA protein, a salmonella effector protein, can promote acetylation of specific host MAPKKs, resulting in suppression of JNK signaling and ultimately hampering the host immune response against salmonella [187]. 

#### 3.5.3. The JNK Pathway in Fungal and Parasitic Infections

The JNK pathway is activated upon fungal infection [61]. For example, JNK in human bronchial alveolar epithelial cells is activated by the virulence factor gliotoxin when infected with the mold *Aspergillus fumigatus*. Then the activated JNK mediates the apoptosis of human bronchial alveolar epithelial cells, ultimately causing invasive aspergillosis [188]. Additionally, the JNK pathway is also activated in host antifungal responses. An example is that *Candida albicans*-activated JNK1 signaling has a negative effect on antifungal innate immunity [189]. 

Similar to bacterial or fungal infections, JNK can be activated by parasitic infections. For example, JNK2 is activated by *Toxoplasma gondii* during the infection process [190]. Then, the activated JNK2 plays a dual role in host resistance and immunity [191].

### 3.6. The JNK Pathway in Other Diseases

JNK also functions in other diseases. Here, we briefly discuss its role in other diseases that have not been mentioned above.

#### 3.6.1. Hearing Loss

JNK is reported to be involved in deafness via promoting apoptosis [26]. Eshraghi et al. found that the JNK pathway is activated in hair cells under external stresses such as cochlear implantation trauma and noise. The activated JNK then induces hair cell apoptosis, ultimately causing deafness [192]. 

#### 3.6.2. Ischemic/Reperfusion Injury

Ischemia/reperfusion injury often occurs in the brain, liver, heart, and kidneys. Although reperfusion is necessary for survival, it will result in tissue injury and activate inflammatory signaling, ultimately promoting JNK activation [36]. Taking myocardial infarction as an example, inflammatory signaling is stimulated in ischemic or necrotic myocardial cells, which ultimately leads to ventricular remodeling. This process involves ASK1-JNK1/2 signaling because ASK1 and JNK1/2 KO mice display reduced cardiac remodeling [36,193,194]. 

#### 3.6.3. Cardiac Hypertrophy

Cardiac hypertrophy is defined as the thickening of the heart muscle under a variety of pathological stresses, such as mechanical stress and inflammation [36]. Although transient enlargement of the heart is a normal physiological phenomenon to adapt to these stimulations, the heart will suffer maladaptive hypertrophy under sustained pathological stresses, ultimately leading to heart failure [195]. The process of cardiac hypertrophy involves the activation of JNK signaling. For example, p-JNK is detected in patients with heart hypertrophy; therefore, JNK signaling is thought to promote myocyte growth, which marks pathologic hypertrophy [36].

#### 3.6.4. Abdominal Aortic Aneurysms

Abdominal aortic aneurysms (AAAs) are one kind of arterial aneurysm. Chronic inflammation and vascular smooth muscle hypertrophy are considered the main reasons for AAA [36]. AAA formation involves JNK activation, as p-JNK is widely expressed in aneurysm tissue from patients with AAA [196]. Moreover, JNK expression in aneurysm tissue affects extracellular matrix metabolism, which further promotes the progression of AAA. Interestingly, pharmacological JNK inhibition alleviates AAA symptoms [196]. 

#### 3.6.5. Renal Fibrosis

Renal fibrosis mainly refers to glomerular fibrosis and tubulointerstitial fibrosis, which ultimately develop into end-stage renal failure. These two processes involve activation of JNK signaling, as phosphorylated JNK is widely observed in glomerular cells and renal tissue from patients with acute and chronic renal injury [197]. Importantly, JNK inhibitors can ameliorate renal fibrosis. A key mechanism is that JNK can phosphorylate SMAD3, thus promoting the expression of pro-fibrotic genes, which further aggravate renal fibrosis [197]. 

#### 3.6.6. Autosomal Dominant Polycystic Kidney Disease

Polycystic kidney disease (PKD) is a degenerative kidney disease wherein the renal tubules are filled with multiple fluid-filled cysts [198]. Autosomal dominant polycystic kidney disease (ADPKD) is one kind of PKD. ADPKD has been shown to be caused by mutations of the polycystin-1 (Pkd1) and Pkd2 genes. Smith et al. recently found that the JNK pathway is involved in ADPKD as inhibition of JNK activity reduces cyst growth [198].

## 4. Targeting the JNK Pathway as a Therapeutic Strategy for Treating Human Diseases

Since the deregulated JNK pathway is widely implicated in multiple types of diseases, it is reasonable that pharmacological inhibition of JNK activation or JNK activity is an effective strategy to combat or ameliorate the disease conditions. Currently, JNK inhibitors are used as candidate drugs for ameliorating deregulated JNK signaling-related diseases. Here, we briefly summarize the potential applications of JNK inhibitors in treating human diseases.

### 4.1. Overview of JNK Inhibitors

Broadly, JNK inhibitors mainly include the following several types of inhibitors: ATP-competitive inhibitors, ATP non-competitive inhibitors, and small peptide inhibitors. Mechanistically, ATP-competitive inhibitors interact with the hinge region of the ATP binding site. The non-competitive inhibitors do not bind to ATP-binding sites but bind to the catalytic sites. By contrast, small peptide inhibitors regulate JNK kinase activity by interacting with JNK’s scaffold or its upstream/downstream molecules or by changing the subcellular localization of JNK [27]. Additionally, some natural phytochemicals also exert inhibiting effects against JNK activity. 

### 4.2. The Application of Synthetic JNK Inhibitors in Human Diseases

Up to now, there are a variety of JNK inhibitors that have been studied for ameliorating the disease state. These inhibitors and their applications for treating diseases are summarized in Table 1. In this review, we take several classic JNK inhibitors as examples to elucidate the application of these inhibitors in treating human diseases. 

#### 4.2.1. ATP-Competitive Inhibitors in Human Diseases and Their Action Mechanisms

Here, we give a brief review of the application of three classic ATP-competitive inhibitors in human diseases. The ATP-competitive inhibitors that were used for therapeutic intervention mainly include SP600125, CEP-1347, and AS601245. The three JNK inhibitors occupy an ATP-binding site. SP600125 directly inhibits JNK activity, whereas CEP-1347 inhibits MLK to weaken JNK signaling [68,204,205]. SP600125, a compound of anthrapyrazolone, is the most extensively studied ATP-competitive JNK inhibitor [1]. Although it lacks specificity, SP600125 has been shown to have therapeutic potential. SP600125 was used to study recovery from a number of the deregulated JNK pathway-associated diseases such as ischemia/reperfusion, cancer cell apoptosis, T2D, acute kidney injury, inflammation, viral infections, and sepsis-induced lung injury [27,197,206]. Mechanistically, as a reversible ATP-competitive inhibitor, SP600125 competitively interacts with the ATP binding site of JNK, leading to a reduction of JNK phosphorylation (reduction of JNK activity), either repressing nuclear transcription factors-mediated transcription of target genes or inhibiting phosphorylation of non-nuclear proteins [207]. Because of the important roles of these proteins in cell growth, proliferation, apoptosis, and metabolic stress, SP600125 can ameliorate those above-mentioned diseases. Additionally, SP600125 has been shown to inhibit the expression of inflammatory genes such as *COX-2*, *IL-2*, *IFN-γ*, and *TNF-α*, so it can also be used to improve inflammation [207]. As mentioned above, different diseases involve different JNK isoforms. For example, JNK3 is mainly involved in CNS-related diseases because of its brain-specificity. Moreover, not all isoforms of JNK play negative roles in disease progression. Thus, the non-specific inhibiting effects of SP600125 on JNK isoforms limit its application. CEP-1347, a semisynthetic derivative of the naturally occurring indolocarbazole (K-252a), has broad serine/threonine and tyrosine (trkA) kinase inhibitory activity and is used in pre-clinical studies for ameliorating Parkinson’s disease [68,208]. In terms of the action mechanism, CEP-1347 occupies an ATP-binding site of MLK, the MAP3K upstream of JNK. Binding of the ATP-binding site prevents phosphorylation and activation of MLK, thus negatively regulating JNK signaling [68]. AS601245, also known as bentamapimod, directly inhibits JNK3 at the ATP-binding site, resulting in inhibition of JNK3 activity [68]. Due to the role of JNK3 in neuronal death, AS601245 displays a significant neuroprotective effect against ischemia-induced neuronal cell death and cancer [1]. Additionally, it also exerted anti-inflammatory activity against rheumatoid arthritis [209].

#### 4.2.2. ATP-Non-competitive Inhibitors in Human Diseases and Their Action Mechanisms

Next, we take several ATP-non-competitive JNK inhibitors (BI-78D3, BI87G3, and compound **9**) as examples to elucidate the application of this type of inhibitor. BI-78D3 is the first ATP-non-competitive JNK inhibitor that was developed by Pellecchia and colleagues. Mechanistically, BI-78D3 inhibits JNK activity by interfering with the interaction of JNK1 and the JIP-derived peptide [29,199]. Functionally, BI-78D3 exerts protective effects against liver damage induced by concanavalin A [199]. Also, BI-78D3 was shown to restore insulin sensitivity [199]. Based on the structure–activity relationship, researchers have developed several JNK inhibitors. These inhibitors have similar action mechanisms; that is, they inhibit JNK signaling by blocking the JNK–JIP interaction [82,83]. For example, BI-87G3, a benzothiazole derivative, inhibits JNK activity by competing with JIP for binding to JNK [210]. Similarly, the thiadizole compound **9** competes with JIP-derived peptides to bind to JNK, thus inhibiting JNK activity. Compound **9** can enhance insulin sensitivity in a T2D murine model [211]. In addition, two types of JNK inhibitors were developed: JIP site inhibitors that target the JIP–JNK interaction and dual site inhibitors that not only target the JIP/JNK site but also disrupt the interaction of ATP and JNK. However, the therapeutic potential of these compounds is unknown [29].

#### 4.2.3. Small Peptide Inhibitors in Human Diseases and Their Action Mechanisms

As discussed above, both ATP-competitive and ATP-non-competitive inhibitors have possible off-target effects. Small peptide inhibitors may provide a more targeted inhibition as they specifically target the JNK binding domain (JBD) or regulatory region [27]. One well-known JNK peptide inhibitor is the D-JNKI-1 peptide (also known as XG-102 or AM-111), which is synthesized according to the JIP and HIV TAT sequences. As for the action mechanism, the D-JNKI-1 peptide works by preventing the interaction of JNK with its JBD-dependent targets. For instance, D-JNKI-1 peptide prevents the activity of MKK4 and MKK7, the two JNK upstream MAPKKs, by interacting with the JBD homology domain [212]. D-JNKI-1 can penetrate the cells. D-JNKI-1 has protective effects against nerve injury [204] and can improve lipid metabolism and ameliorate T2D conditions [213]. Recently, D-JNKI-1 has been shown to have beneficial effects on inflammation, pain post-surgery, and hearing loss [27,68,200]. Islet Brain (IB1/2) peptide, another well-known JNK peptide inhibitor, contains a similar JBD domain. Mechanistically, the IB1/2 peptide resembles JNK-interacting protein (JIP) to inhibit JNK. IB1/2 can improve insulin resistance and glucose tolerance [214]. Other peptide inhibitors, whose structures are similar to those of JIP, all exert neuroprotective effects [215].

#### 4.2.4. Dual ATP and Substrate-Competitive Kinase Inhibitors

Notably, designing dual ATP and substrate-competitive kinase inhibitors is a useful approach to exploring novel JNK inhibitors. Stebbins et al. used this strategy to design JNK inhibitors and obtained a bidentate compound (compound **19**) that showed a beneficial effect against diabetes. Compound **19** inhibited JNK activity by concurrently targeting the ATP-binding site and substrate-binding sites [216].

### 4.3. The Application of Natural Phytochemicals That Inhibit JNK Activity in Human Diseases

Many natural phytochemicals showed inhibitory effects on JNK activity and were considered candidate drugs for therapeutic intervention against JNK-related diseases. For instance, the C66 natural variant of curcumin can alleviate diabetes and its complications by inhibiting JNK [27,201,202,203]. Mechanistically, C66 ameliorates diabetes-induced pathological changes by inhibiting JNK2 activity and upregulating Nrf2 expression [201]. Leupeol, a natural pentacyclic triterpene, can inhibit JNK activity. Previous studies have shown that leupeol has anti-hyperglycemic and anti-dyslipidemic activity and has a protective effect against lipopolysaccharide (LPS)-induced neuro-inflammation [217,218]. Leupeol inhibits LPS-induced neuroinflammation by inhibiting phosphorylation of JNK and decreasing the generation of pro-inflammatory cytokines, including TNF-α, iNOS, and IL-1β [217]. Gingerol can modulate lipid metabolism by inhibiting TNF-α-induced JNK activation [219]. These studies demonstrate that plant-based natural products that inhibit JNK activity also have therapeutic potential for human diseases.

### 4.4. The Challenges We Faced When We Were Developing JNK Inhibitors

Although small-molecule inhibitors of JNKs show beneficial efficacy against various JNK-related diseases to some extent, it should be noted that the non-specificity of these inhibitors should be considered. That is, when developing JNK inhibitors, off-target effects and potential side effects need to be considered. Here, in order to emphasize the limitations of current JNK inhibitors and help develop new JNK inhibitors in the future, we provide brief information about the (isoform) specificity and off-target effects of several above-mentioned JNK inhibitors, as well as the structural information of JNKs complexed with their corresponding inhibitors (Table 2).

## 5. Conclusions and Perspectives

The JNK signaling pathway regulates a variety of cell behaviors, such as cell proliferation, differentiation, and cell survival. Dysregulated JNK signaling is implicated in various human diseases such as metabolic disorders (e.g., obesity, T2D, and NAFLD), neurological disease (e.g., neurodegenerative disease), chronic inflammatory disease (e.g., RA and IBD), infectious disease, and cancer. Although we have made some progress in understanding the critical role of the JNK pathway in human disease, our knowledge about the function of JNK is still limited. Especially several questions, for instance, how the JNK pathway is strictly regulated under physiological and pathophysiological conditions and why the isoform-specific and cell type-specific responses of JNK are important for disease progress, remain to be addressed. Future studies are needed to elucidate the JNK isoform-specific role in a tissue-type-specific manner and to better explore the function of other molecules in the JNK pathway in disease progression.

Since JNKs play important roles in almost all human diseases, a common strategy for treating the disease with the target JNK, theoretically, would be desirable. For example, as described in Table 1, the JNK inhibitor SP600125 can be used to treat several diseases, such as ischemia/reperfusion, cancer, inflammation, viral infections, and sepsis-induced lung injury. However, this would require the identification of common or individual disorders of JNK signaling that occur under the corresponding disease states. Thus, identification of common or individual deregulated JNK signaling may be a future direction.

In terms of developing JNK inhibitors, off-target effects and potential side effects need to be considered. Small-molecule inhibitors such as ATP-competitive inhibitors, ATP non-competitive inhibitors, and small peptide inhibitors are extensively investigated, and some of these inhibitors show beneficial efficacy against various JNK-related diseases. However, avoiding the non-specificity of these inhibitors is still a great challenge. For example, because of the high homology (98% similarity) in the ATP binding pocket among JNK1, JNK2, and JNK3, it is a challenge to explore isoform-selective inhibitors. Therefore, it is urgent and necessary for us to identify substrate-specific or JNK isoform-specific inhibitors. In the future, targeting JNK kinase regulatory sites may be a new approach for discovering JNK isoform-specific inhibitors. In addition, because JNK signaling sometimes plays a conflicting role in some diseases, we need to consider and balance the advantages and side effects of JNK inhibitors. Thus, caution is still needed for the use of these JNK inhibitors in future clinical applications.

## Figures and Tables

**Figure 1 biomolecules-14-00243-f001:**
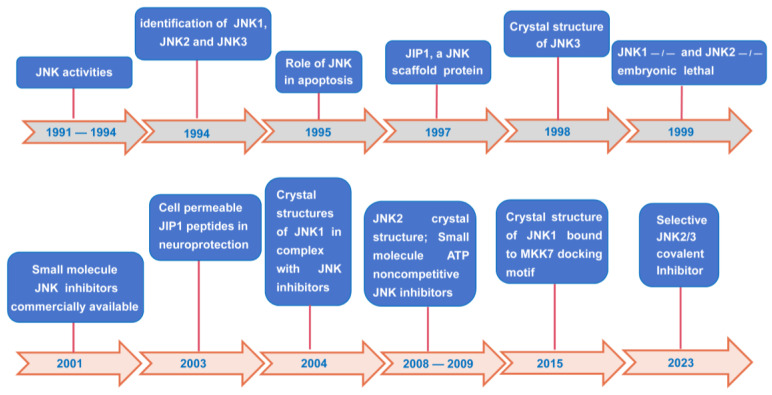
**Landmark discoveries in the area of JNK research.** The timeline of significant discoveries spans from the initial identification of JNKs as stress-activated protein kinases to the discoveries of selective JNK isoform-specific inhibitors.

**Figure 2 biomolecules-14-00243-f002:**
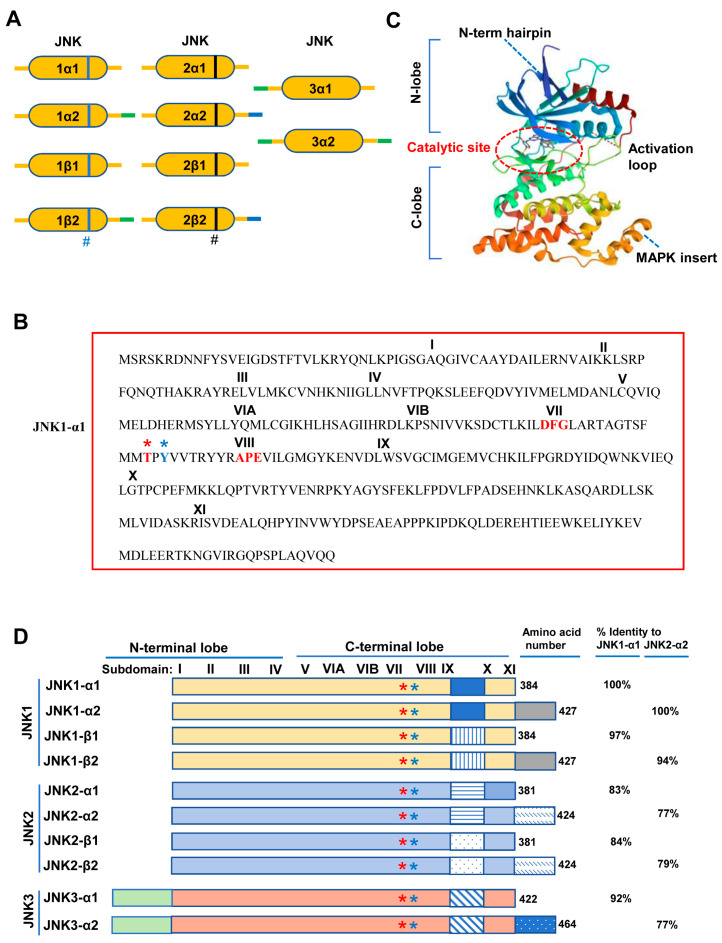
**Alternative splicing isoforms of the JNK MAPKs.** (**A**) Characteristics of the alternative splicing isoforms of JNK1, JNK2, and JNK3. # indicates alternative exon 6 between the α and β isoforms of JNK1 (blue symbol “#”) or JNK2 (black symbol “#”) that result in a change of 5–7 amino acids in this region. Each of these JNK kinases is composed of a kinase domain (central region) as well as N- and C-termini represented by a line on the left (N terminus) and right (C terminus) of all kinase domains. Different colors and lengths in the N- or C-terminus represent distinct sequences and number of AAs compared to the main isoform. (**B**) The amino acid sequence of JNK1-α1, which contains 384 amino acids, is shown as a typical member of this family. JNK1 contains all 11 protein kinase subdomains as indicated (I–XI). The protein kinase activation loop is located between domains VII and VIII, which contains the conserved amino acid motifs DFG (red) and APE (red). The threonine (T, red) and tyrosine (Y, red) are within this loop and indicated by a red asterisk and a blue asterisk, respectively. (**C**) The structure of the human JNK1 protein (PDB id: 3ELJ). The N-/C-terminal lobes, N-terminal hairpin, and MAPK insert site as well as the activation loop are indicated. (**D**) Protein structural diagram of ten different isoforms of JNK produced by the alternative splicing of three JNK genes (JNK1, JNK2, and JNK3). The differences are indicated by the patterned rectangles. The similarity of the shorter forms is analyzed by comparing the percent identity of JNK1-β1, JNK2-α1, JNK2-β1, and JNK3-α1 with JNK1-α1. Similarly, the longer forms can be compared, and the percent identity of JNK1-α2 with JNK1-β2, JNK2-α2, JNK2-β2, and JNK3-α2 is shown. Red asterisk indicates the threonine, while blue asterisk indicates the tyrosine within the activation loop that is located between domains VII and VIII.

**Figure 3 biomolecules-14-00243-f003:**
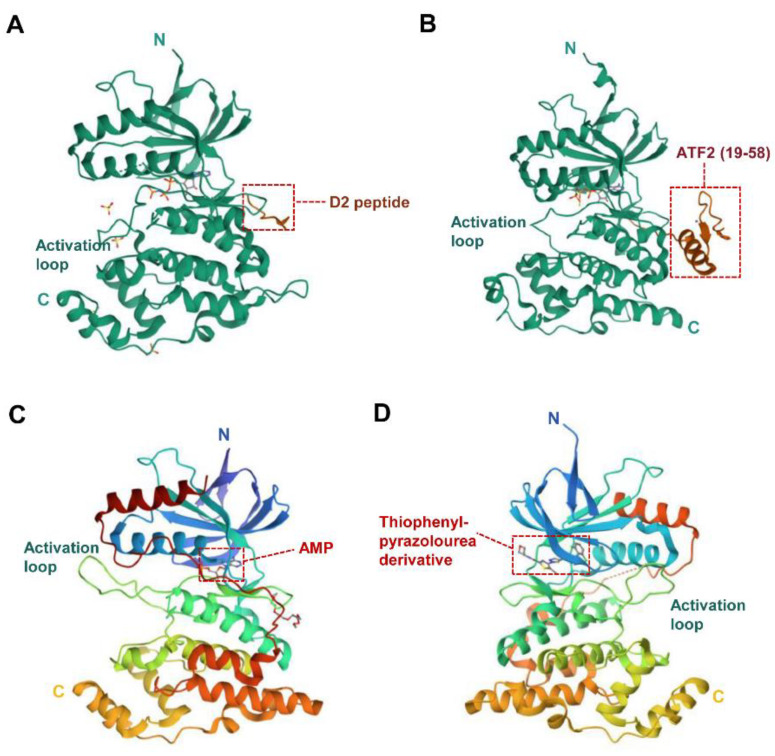
**Crystal structures of JNKs and their ligands.** (**A**) Crystal structure of JNK1 bound to a MKK7 docking motif (PDB ID: 4UX9). The D2 peptide (QRPRPTLQLPLA), which is derived from the D2 docking site of MKK7, is the bound ligand of JNK1. (**B**) Crystal structure of JNK1 in complex with the activating transcription factor 2 (ATF2) fragment (PDB ID: 6ZR5). Cyclic AMP-dependent transcription factor ATF2 (19-58) is the bound ligand of JNK1. (**C**) Crystal structure of AMP-bound human JNK2 (PDB ID: 7N8T). Adenosine monophosphate (AMP) is the bound ligand of JNK2. (**D**) Crystal structure of JNK3 complexed with a thiophenyl–pyrazolourea derivative (PDB ID: 7KSI). The thiophenyl–pyrazolourea derivative is the bound ligand of JNK3. All the four structural images were down-loaded from https://www.rcsb.or/. Access date: 28 October 2023.

**Figure 4 biomolecules-14-00243-f004:**
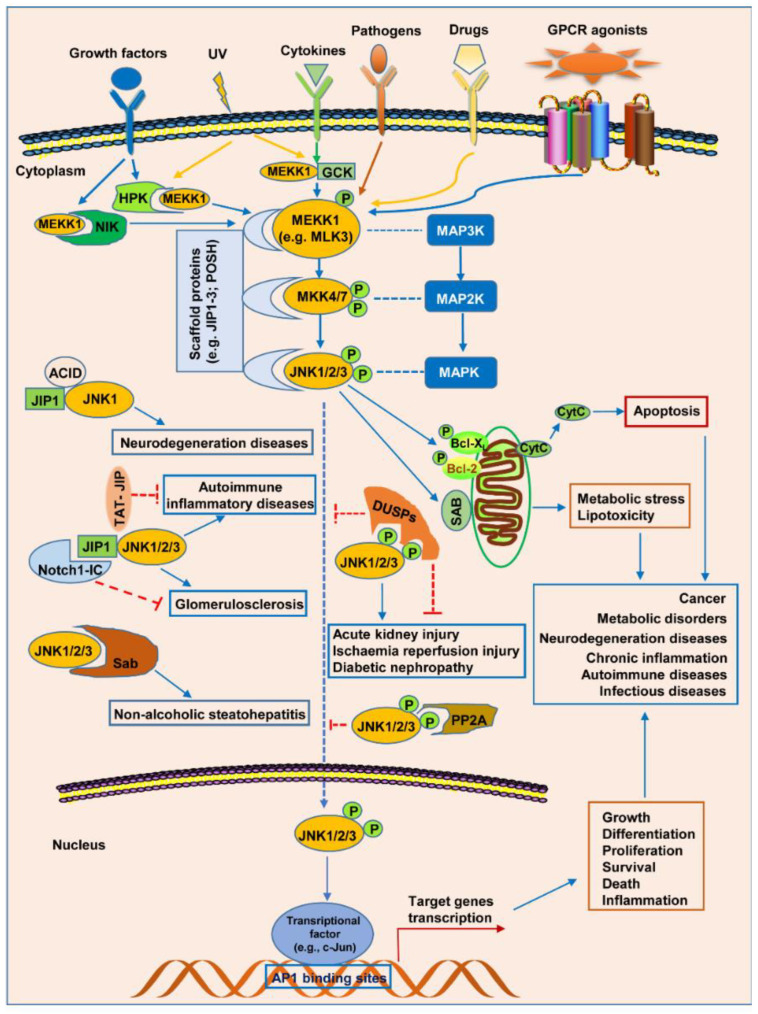
**The general activation mechanism of the JNK signaling pathway.** JNK kinases are activated by a series of phosphorylation events in response to extracellular stimuli such as growth factors, cytokines, environmental stresses, pathogens, GPCR agonists, and drugs. These stimuli first result in activation of the STE20 protein homologue germinal center kinases, including HPK, NIK, and GCK, which interact with and phosphorylate MEKK1, or directly cause the activation of membrane proximal MAP3K kinases such as ASK1, TAK1, or MLK3, which then phosphorylates and activates MKK4 and MKK7. The activated MKK4 or MKK7 further phosphorylates and activates distinct JNK isoforms through dual phosphorylation on tyrosine and threonine residues within a conserved tripeptide motif. Notably, DUSPs and PP2A can dephosphorylate JNK and inhibit JNK signaling. Upon activation, JNK, on the one hand, phosphorylates several nuclear transcription factors such as c-Jun, ATF2, and SP1, causing the transcription of target genes. On the other hand, JNK also interacts with and phosphorylates a number of non-nuclear proteins, including mitochondrial BCL-2 family members (e.g., BCL-XL and BCL-2) and some cytoplasmic proteins. Finally, these proteins mediate a variety of cellular responses, including cell growth, differentiation, proliferation, survival, apoptosis, inflammation, metabolic stress, and cell death. In addition, several protein complexes are also involved in regulating JNK signaling. For instance, the complex composed of ACID, JIP1, and JNK1, the complex composed of JIP1 and JNK1/2/3, and the complex composed of Sab and JNK1/2/3 are also involved in JNK signaling transduction and are critical for the development of neurodegenerative diseases, autoimmune inflammatory diseases/glomerulosclerosis, and non-alcoholic steatohepatitis, respectively. The TAT-JIP1 peptide and Notch 1-IC are used for ameliorating autoimmune inflammatory diseases and glomerulosclerosis, respectively.

**Figure 5 biomolecules-14-00243-f005:**
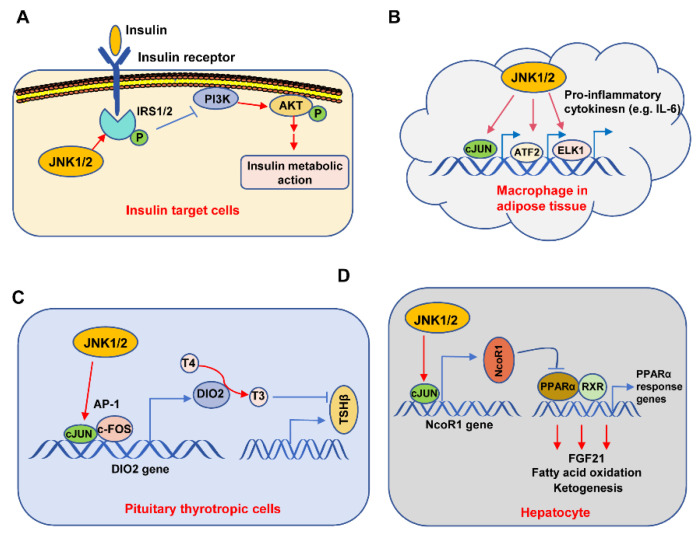
**Molecular mechanisms for the involvement of the JNK pathway in insulin resistance.** The JNK1/2 kinases were important for obesity-driven insulin resistance through four mechanisms: (**A**) In insulin-target cells, JNK1/2 directly phosphorylates IRS1 and IRS2 at serine and threonine residues, resulting in reduced tyrosine phosphorylation of IRS1/2 molecules, which further weakens the response of the PI3K/AKT signaling pathway to insulin. (**B**) In adipose tissue macrophages, JNK1/2 phosphorylates some transcriptional factors such as c-JUN, ATF2, and ELK1, stimulating gene transcription of pro-inflammatory cytokines, leading to increased levels of inflammatory “M1” cytokines (e.g., IL-6), which ultimately drive insulin resistance. (**C**) In pituitary thyrotropic cells, JNK1/2 phosphorylates c-JUN and c-FOS, sustaining the transcription and subsequent expression of DIO2, which mediates the shift from T4 to T3, finally leading to a T3-dependent reduction of TSHβ production. The reduced TSH ultimately causes low circulating levels of thyroid hormone, increases metabolic efficiency, and increases adiposity, finally driving insulin resistance. (**D**) In hepatocytes, JNK1/2 phosphorylates c-JUN, sustaining the transcription and expression of the transcription co-repressor *NcoR1*. Then, NcoR1 represses PPARα-driven gene expression, causing a reduction in FGF21 production, fatty acid oxidation, and ketogenesis, ultimately promoting fatty liver and insulin resistance.

**Figure 6 biomolecules-14-00243-f006:**
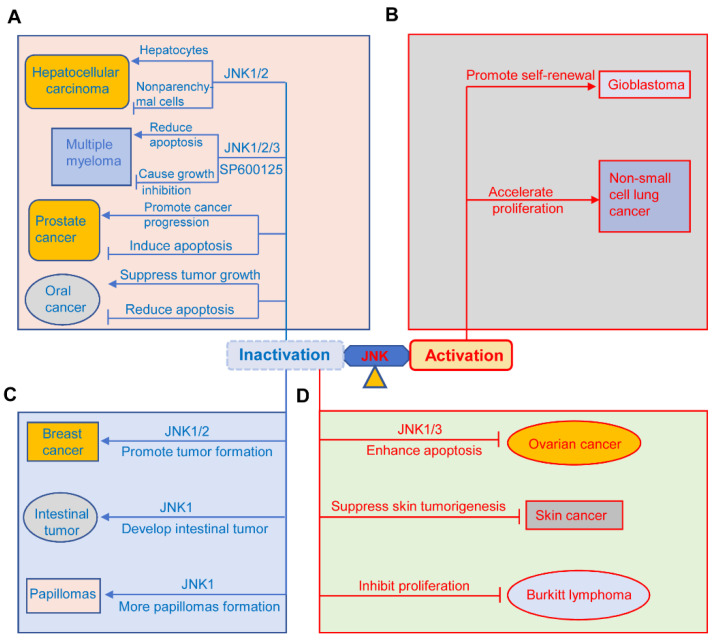
**Pro-oncogenic and anti-oncogenic roles of JNK signaling in cancers.** (**A**) The JNK signaling plays a dual role in the occurrence and progression of different kinds of cancer, including hepatocellular carcinoma, multiple myeloma, prostate cancer, and oral cancer. (**B**) JNK signaling promotes the progression of glioblastoma and non-small-cell lung cancer. (**C**) Deficiency or inhibition of different JNKs promotes tumor formation in breast cancer, intestinal tumors, and papilloma. (**D**) Inhibition of the JNK pathway suppresses the tumorigenesis of ovarian cancer, skin cancer, and Burkitt lymphoma. Activation indicates sustained activation of the JNK pathway; inactivation represents deficiency of JNK1, JNK2, JNK3, or compound deficiency of JNK1 and JNK2; inactivation also indicates inhibition of JNK activity by JNK inhibitors such as SP600125.

**Table 1 biomolecules-14-00243-t001:** Summary of JNK inhibitors and their application in human disease models.

Class	Drug	Application in Disease Model	Reference
ATP-competitive small-molecule inhibitors	SP600125	Ischemia/reperfusion; cancer; T2D; acute kidney injury; inflammation; viral infections; and sepsis-induced lung injury	[27,181,185]
	CEP-1347	parkinson’s disease and cancer	[41,45]
	AS601245 (bentamapimod)	Ischemia-caused neuronal damage; cancer; and rheumatoid arthritis (RA)	[1,186]
	SR-3576	Neurodegenerative diseases	[28]
	CC-930	Renal/liver fibrosis	[41]
	CC-401	Renal/liver injury	[41]
ATP-non-competitive small-molecule inhibitors	BI-78D3	Liver damage and type 2 diabetes	[29,199]
	Compound **9**	Type 2 diabetes	[189]
Small peptide inhibitors	D-JNKI-1 (XG-102 or AM-111)	Neuron damage; lipid metabolism; type 2 diabetes; cerebral ischemia; hearing loss; hypoxia-induced retinopathy; liver injury; and acute inflammatory insult	[27,68,190,191,200]
	IB1/2	Type 2 diabetes	[190,193]
Natural phytochemicals	C66	Diabetes; aortic damage; and atherosclerosis	[27,201,202,203]
	Lupeol	Neuroinflammation; diabetes; and lipid metabolism	[198,199]
	Gingerol	Lipid metabolism	[200]
	Capsaicin	Inflammatory insult	[26]

**Table 2 biomolecules-14-00243-t002:** The specificity and structural information of several JNK inhibitors.

Inhibitor	Specificity	Off-Target Effects	Structure of JNK Complexed with Inhibitor (PDB ID)
SP600125	JNK1/2/3	Yes	1UKH
CEP-1347	MLK	Yes	Not reported
AS601245	JNK1/2	Yes	Not reported
SR-3537	JNK3	Yes	3FI3
SR-3576	JNK3	Not reported	Not reported
CC-359	JNK1/2/3	Not reported	3TTJ
CC-930	JNK1/2/3	Not reported	3TTI
BI-78D3	JNK/JIP1	Not reported	Not reported
D-JNKI-1	JNK1	Not reported	Not reported
JIP1 peptide	JNK1/3	Not reported	4H39; 1UKI
C66	JNK2	Yes	Not reported
Lupeol	JNK1	Yes	Not reported
Gingerol	JNK1/2	Yes	Not reported
Capsaicin	JNK1	Yes	Not reported

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
