# Peer review of "The Role of the Dysregulated JNK Signaling Pathway in the Pathogenesis of Human Diseases and Its Potential Therapeutic Strategies: A Comprehensive Review"

_biomolecules, 2024, doi:10.3390/biom14020243_

Round 1

Reviewer 1 Report

Comments and Suggestions for Authors

Interesting review! Mechanistical aspects of this paper can be improved by including a section of protein complex as a key mechanism for JNK signaling in signaling and translational medicine via diagram.  

Author Response

A1. We would like sincerely to thank the reviewer for his/her professional suggestion. After carefully reviewing the literatures, we have added  several protein complexes, which are also involved in JNK signaling transduction and the development of some diseases, in the revised figure 4 (previous figure 3). We also marked the functions of these protein complexes in the development of diseases via the diagram in this revised figure 4. For instances, the complex composed of ACID, JIP1 and JNK1, the complex composed of JIP1 and JNK1/2/3, and the complex composed of Sab and JNK1/2/3 are also involved in JNK signaling transduction and critical for neurodegeneration diseases, autoimmune inflammatory diseases/glomerulosclerosis and non-alcoholic steatohepatitis, respectively. TAT-JIP peptide and Notch 1-IC (Notch intracellular domain) can be used for ameliorating autoimmune inflammatory diseases and glomerulosclerosis, respectively, through a competitive inhibition of the interaction of JIP1 and JNK1/2/3. Correspondingly, we added the description about these protein complexes and their roles in JNK signaling pathway and diseases in both the figure 4 legend and 2.4 section (General activation mechanism of JNK signaling pathway) in the text. The corresponding revisions were highlighted by underlining and in red color in this revision of our manuscript (the last paragraph, page 6; paragraph 1, page 7).

Reviewer 2 Report

Comments and Suggestions for Authors

The authors have written an extensive review of altered JNK signalling found in a range of disease states in humans.  Having read the manuscript I have the following comments:

1.  Delete the sentence "In general...." for L51 as it is repeating that stated on L47.

2. L178 replace "secret" with "secrete"

3. Section 3.4.1 remove the abbreviation MM and write multiple myeloma instead.  MM also is an abbreviation for malignant melanoma.

4. L510 the word escherichia coli should be written in italics and Escherichia should have a capital E.

5. L520 Aspergillus fumigatus should be written in italics, please correct.

6. L523 Candida albicans should be written in italics, please correct.

7. L526 Toxoplasma gondii should be written in italics, please correct.

8. Section 4 on the application of therapeutic inhibitors should be extended to further discuss the mechanism by which these compounds inhibit JNK signalling in different disease states.

Author Response

Q1. “The authors have written an extensive review of altered JNK signalling found in a range of disease states in humans.  Having read the manuscript I have the following comments:

  1.  Delete the sentence "In general...." for L51 as it is repeating that stated on L47.”

A1. We would like sincerely to thank the reviewer for his/her comments. According to the comment above, we have  deleted the sentence "In general...." for L51 in this revision of our manuscript (paragraph 2, page 2). The corresponding revisions were highlighted by underlining and in red color in this revision of our manuscript.

Q2. “L178 replace "secret" with "secrete"”

A2. We would like sincerely to thank the reviewer for his/her comments. According to the comment above, we have replaced the word "secret" in previous L178 with “secrete" in line 399 in this revision of our manuscript. The corresponding revisions were highlighted by underlining and in red color in this revision of our manuscript.

Q3.“Section 3.4.1 remove the abbreviation MM and write multiple myeloma instead.  MM also is an abbreviation for malignant melanoma.”

A3. We would like sincerely to thank the reviewer for his/her helpful comments. According to the comment above,  we have removed the abbreviation MM and used “multiple myeloma” to replace it in section 3.4.1 in this revision of our manuscript. The corresponding revisions were highlighted by underlining and in red color in this revision of the manuscript (paragraph 2, page 16).

Q4.“L510 the word escherichia coli should be written in italics and Escherichia should have a capital E.”

A4. We would like sincerely to thank the reviewer for his/her professional comments.  According to the comment above,  we have written the word escherichia coli in italics and used a capital E to replace the lowercase E  in the word “Escherichia” in section 3.5.2 in this revision of our manuscript. The corresponding revisions were highlighted by underlining and in red color in this revision of the manuscript (line 682, paragraph 5, page 17).

Q5. “L520 Aspergillus fumigatus should be written in italics, please correct.”

A5. We would like sincerely to thank the reviewer for his/her constructive suggestions. According to the comment above,  we have written the word Aspergillus fumigatus in italics in section 3.5.3 in this revision of our manuscript. The corresponding revisions were highlighted by underlining and in red color in this revision of the manuscript (line 692, paragraph 1, page 18).

 Q6. “L523 Candida albicans should be written in italics, please correct.”

A6. We would like sincerely to thank the reviewer for his/her comments. According to the comments above, we have written the word  Candida albicans in italics in section 3.5.3 in this revision of our manuscript. The corresponding revisions were highlighted by underlining and in red color in this revision of the manuscript (line 695, paragraph 1, page 18).

Q7. “L526 Toxoplasma gondii should be written in italics, please correct.”

A7. We would like sincerely to thank the reviewer for his/her comments. According to the comments above, we have written the word “Toxoplasma gondii”  in italics in section 3.5.3 in this revision of our manuscript. The corresponding revisions were highlighted by underlining and in red color in this revision of the manuscript (line 698, paragraph 2, page 18).

Q8.“Section 4 on the application of therapeutic inhibitors should be extended to further discuss the mechanism by which these compounds inhibit JNK signalling in different disease states.”

A8. We would like sincerely to thank the reviewer for his/her comments. According to the comments above, we have further discussed the specific mechanisms by which these therapeutic compounds inhibit JNK signalling in different disease states. These JNK inhibitors mainly include ATP-competitive inhibitors (SP600125, CEP-1347 and AS601245), ATP-noncompetitive inhibitors (BI-78D3, BI87G3 and compound 9),  small peptide inhibitors (D-JNKI-1 and IB1/2 peptide),  dual ATP and substrate competitive kinase inhibitor (bidentate compound 19) and natural phytochemicals (C66 and Leupeol). In addition, we also added four subtitles (4.2.1...; 4.2.2…; 4.2.3…; 4.2.4) in the section 4.2 according to the classification of JNK inhibitors. For example, 4.2.1. ATP-competitive inhibitors in human diseases and their action mechanisms. The corresponding revisions were highlighted by underlining and in red color in section 4 in this revision of the manuscript ( from  page 20 to page 22).

Reviewer 3 Report

Comments and Suggestions for Authors

A review on JNKs' signaling pathways in human disease would be very welcome. This review is an excellent collection of published papers on JNKs in various diseases. However, a review is required to bring together different observations at the molecular level and gain insights into common ways in which an enzyme is regulated and how it influences downstream signaling pathways in the cell. In the case of deviations in the common pathways of activation and inactivation, new questions arise. Unfortunately, this manuscript does not fulfill the requirements for a good review. It is very superficial and most of the paragraphs end with the sentence " The data show that a dysregulated JNK signaling plays a role in the disease ......” Such an overview article could also be summarized in a table. It is a simple sequence of statements without any critical reflection. In addition, there is no description of molecular mechanisms. It remains to be seen which molecular functions JNKs fulfill. There is a lack of information on how JNKs are regulated. Are there common targets for JNKs? Are there tissue-specific expressions? Are there commonalities? What does the statement mean that the JNK signaling pathway is disturbed? What causes the disturbance? Are there mutations that lead to the activation of JNKs? Is the disorder due to an altered amount of JNKs? How are JNKs regulated, e.g. by increasing or decreasing the level of JNKs, or by activating or inactivating kinase activity? Sometimes, the authors use the term “JNK”, sometimes they use one or all three isoforms of JNK. This sounds as if all JNK isoforms have the same properties and functions. How do the different isoforms of JNK differ in terms of genome organization and protein sequences? How do the isoforms differ in their functions and properties. Are there structural differences? Are they expressed in different tissues?  

 Which activation or inactivation mechanisms for JNKs do exist? Is there a tissue-specific expression? Do the JNKs occur in all cellular compartments? The authors present various inhibitors. Do they inhibit the kinase activity, the subcellular localization or the interaction with other cellular binding partners? Which activation or inactivation mechanisms exist? Since JNKs play a role in almost all human diseases based on the lists in this manuscript, a common strategy for treating the disease with the target JNK would be desirable based on the data. However, this would require the identification of common or individual disorders of JNK signaling. Unfortunately, this has not been done.

The manuscript also contains a number of formal errors, both in the illustrations and in the text. The English grammar needs to be revised by a native English speaker.

 The following points are not a complete list of weaknesses, but serve only to illustrate some flaws.

 Lines 47 and 53 are a duplication.

In Fig. 1 B, the information on 2004 must be reformulated. The time line is a continuum and can therefore not be A and B. In Fig. 1 B, the information on 2004 must be reformulated

Row 44 and rows 84- 85 are repetitions

Line 102: Which kinase is meant?

Legend of Fig. 2 is incomplete. Information about the bound ligands and an indication in the figure is necessary.

Lines 117-119: What is known about the activation of JNKs by viral proteins?

Fig. 3: The MAP3K, MAP2K and MAP3K line is not integrated in the Fig. Is the numbering correct?

Figure 3 suggests a pure signaling cascade. Are there other kinases or other cellular or viral proteins that phosphorylate or bind to members of the signaling cascade shown?

Lines 173-175: a citation is missing

A protein phosphatase is mentioned in the text, but not included in the figure. Are there other phosphatases that interfere with the signaling cascade?

Figure 4 A-D is systematically wrong. Only IRS1/2 phosphorylation is shown, Akt phosphorylation is missing. What means “Insulin action? In B- D transcription or transcribed genes are shown. The text in lines 197 - 208 is only incompletely reflected in Fig. 4, if at all.

Lines 268- 269: It is not clear which function of JNK plays an important role.

Line 293: What is meant by "over-activation of JNK pathway..."?  Is the kinase activity activated or is there an elevated level of JNKs, or both?

What means  “ The JNK signaling in CNS… is stronger”?

Lines 324 “JNK also mediates PD progression..” Does JNK this alone or together with other factors? What is the molecular basis for this mediation?

Lines 345- 350 there is paper cited which showed an upregulation of phosphorylated JNK. This is an indication that not only JNK functions as a kinase but it is itself phosphorylated by one or more protein kinases. The kinases which phosphorylate JNKs would give an important inside view into an upstream regulation of JNKs.

Figure 5 would be a good summary but the authors must explain what activation and inactivation means.

Line 605: “ .. as mentioned above, different diseases involve different JNK isoforms… “. This is a very interesting point which is not consequently addressed in the review.

With regard to inhibitors, information about the specificity, isoform specificity, off target effects etc. would be very helpful. Are there any crystal structures known for JNKs alone or the complex with inhibitors?

Comments on the Quality of English Language

Moderate editing of English language is required

Author Response

Reviewer 3

Q1. “A review on JNKs' signaling pathways in human disease would be very welcome. This review is an excellent collection of published papers on JNKs in various diseases. However, a review is required to bring together different observations at the molecular level and gain insights into common ways in which an enzyme is regulated and how it influences downstream signaling pathways in the cell. In the case of deviations in the common pathways of activation and inactivation, new questions arise. Unfortunately, this manuscript does not fulfill the requirements for a good review. It is very superficial and most of the paragraphs end with the sentence " The data show that a dysregulated JNK signaling plays a role in the disease ......” Such an overview article could also be summarized in a table. It is a simple sequence of statements without any critical reflection. In addition, there is no description of molecular mechanisms. It remains to be seen which molecular functions JNKs fulfill. There is a lack of information on how JNKs are regulated. Are there common targets for JNKs? Are there tissue-specific expressions? Are there commonalities? What does the statement mean that the JNK signaling pathway is disturbed? What causes the disturbance? Are there mutations that lead to the activation of JNKs? Is the disorder due to an altered amount of JNKs? How are JNKs regulated, e.g. by increasing or decreasing the level of JNKs, or by activating or inactivating kinase activity? Sometimes, the authors use the term “JNK”, sometimes they use one or all three isoforms of JNK. This sounds as if all JNK isoforms have the same properties and functions. How do the different isoforms of JNK differ in terms of genome organization and protein sequences? How do the isoforms differ in their functions and properties. Are there structural differences? Are they expressed in different tissues?   ”

A1. We would like sincerely to thank the reviewer for his/her comments. We will successively explain the above-mentioned issues as follows:

       We elucidated the general molecular mechanisms of JNK activation  in the 2.4 section in our previous manuscript. In this revision, we have broadened and deepened the molecular mechanism according to your and another reviewer’s comments, and summarized the molecular mechanism in a revised figure 4  (previous figure 3).  Additionally, we have further revised the description about the molecular mechanism of JNK activation according to this revised figure 4 in this revision. We think that this new figure 4 can provide a good summary of the molecular functions of JNK and offer the information on how JNKs are regulated. Correspondingly, we have added the description about the regulation and molecular functions of JNKs in this revion of the manuscript (from page 5 to page 8).

It is well known that once activated, JNK MAPKs can phosphorylate their downstream target proteins such as ATF, c-JUN, etc. Importantly, c-Jun is an exclusive JNK MAPK substrate [Int J Biochem Cell Biol. 2001 Nov;33(11):1047-63]. However, JNKs can also phosphorylate other transcription factors, for example, ATF2. ATF2 is a JNK MAPK substrate that heterodimerises with c-Jun and stimulates expression of the c-jun gene [Int J Biochem Cell Biol. 2001 Nov;33(11):1047-63]. Therefore, through activation of both c-Jun and ATF2, JNK can regulate the abundance and activity of c-Jun. Elk-1 is another direct JNK MAPK target, whose product forms the AP-1 heterodimer with c-Jun. In the cases of the transcription factors ATF2 and Elk-1, it should also be noted that other protein kinases can lead to their phosphorylation and activation. Specifically, ATF2 can also be phosphorylated by p38 MAPKs whereas Elk-1 may also be phosphorylated by both ERK and p38 MAPKs [Int J Biochem Cell Biol. 2001 Nov;33(11):1047-63]. Thus, there is a possibility of cross-talk between the MAPK pathways at the level of their transcription factor substrates. These findings imply that c-Jun may be the common target for JNKs. Actually, either this revised figure 4 or previous figure 3 clearly illustrates that JNK1, JNK2 and JNK3 all can phosphorylate c-Jun. 

In terms of tissue-specific expression of JNK, we described about the tissue-specific expressions of JNKs in our previous manuscript and this revision of  our manuscript in “2.2 section” (paragraph 2, page 4), i.e., JNK1/2 expresses in any one cell and in various tissues throughout the body, whereas JNK3 mainly expresses in a few tissues including brain, testis, heart and pancreas. We think that the commonalities of JNKs include, but not limited to, the following elements: (1). Different JNK isoforms have a similar structure (e.g. N-lobe, c-lobe and activation loop);  (2). The JNK isoforms all belong to typical serine/threonine protein kinase containing all 11 subdomains; (3). All of the JNK isoforms can phosphorylate c-Jun; (4). They can mediate cell survival or cell death; (4). JNK isoforms can be negatively regulated by DUSPs or PP2A, and regulated by JIP. In fact, although we did not explicitly summarize these commonalities, we have described them throughout the previous manuscript. In this review, we summarized these commonalities in “2.4 General activation mechanism of JNK signaling pathway” section (paragraph 2, page 8). 

As we know, JNKs belong to serine/threonine protein kinase family. As an enzyme, the kinase activity of JNK is strictly regulated in the cell. The sentence “ the JNK signaling pathway is disturbed” means that the kinase activity is in the case of deviations in the common pathways of activation and inactivation. That is, the JNK signaling pathway become deregulated. The JNK signaling pathway can be disturbed when cells are stimulated by external factors including cytokines, growth factor, ROS, heat shock, shear stress, pathogens, and drugs (the last paragraph, page 5). In addition, abnormal expressions of mutations of some genes in JNK pathway in cells can disturb JNK signaling. For instances,  the constitutively active mutant of MKK4 can cause JNK sustained activation. By contrast, DUPS overactivated in cells can lead to inactivation of JNK.

To our knowledge,  no reports have shown that  mutations of JNKs themselves lead to activation of JNKs, however, the constitutively active mutations of some upstream kinases (e.g. TAK1 or MKK4/7) or other molecules (e.g. agouti-related peptide  can activate JNKs (AgRP) [J Biol Chem. 1998, 273(10):5423-6; Cell Rep. 2014;9(4):1495-506.].     

In general, JNKs are regulated by activating or inactivating kinase activity. As for the properties and functions of JNK isoforms,  JNK isoforms have similar structures and functions to mediate JNK signaling, for example, they can phosphorylate transcription factor such as ATF, c-JUN, etc. However, the isoforms of JNK have different substrate specificity, thus they can phosphorylate different non-nuclear substrates or downstream nuclear transcription factors, indicating that different isoforms of JNK have different functions. Similar to our opinion, it is thought that the main JNK MAPKs components in JNK pathway, which are very similar to each other, funnel the signals towards their downstream targets without allowing much fluctuation. Nevertheless, in some cases, even minor sequence differences between these main components are sufficient to cause different activities [Cells. 2021 Dec 8;10(12):3466.]. Therefore, we think that JNK1-3 are somewhat different, as each gene has several alternatively spliced protein products, which varies between cell lines though they have  similar structures and functions.

As an evolutionarily-conserved family of serine/threonine protein kinases, the sequences of the JNK genes are highly conserved through evolution [J Mol Biol, 2020 Mar 27;432(7):2121-2140;. Alternative gene splicing is now recognised as a mechanism to produce different splicing forms [Int J Biochem Cell Biol. 2001 Nov;33(11):1047-63]. The genome organization, protein sequences, typical  structure and structural differences of JNKs are summarized in a new figure 2 in this revision of our manuscript. In addition, as for the expression localization, we previously described that JNK1/2 expresses in any one cell and in various tissues throughout the body, whereas JNK3 mainly expresses in a few tissues including brain, testis, heart and pancreas (paragraph 2, page 4). Due to the differences in tissue expression, different JNK isoforms may pay a different role in different diseases progress.

Overall, to address the above issues, we proved a new figure 2 and broadened and deepened the action mechanism, and broaded the mechanism via diagram  in this revised figure 4.  And we have added the corresponding descriptions  in this revision of our manuscript. The corresponding revisions in section 2 were highlighted by underlining and in red color in this revision of the manuscript (from page 3 to page 8).

Q2. “Which activation or inactivation mechanisms for JNKs do exist? Is there a tissue-specific expression? Do the JNKs occur in all cellular compartments? The authors present various inhibitors. Do they inhibit the kinase activity, the subcellular localization or the interaction with other cellular binding partners? Which activation or inactivation mechanisms exist? Since JNKs play a role in almost all human diseases based on the lists in this manuscript, a common strategy for treating the disease with the target JNK would be desirable based on the data. However, this would require the identification of common or individual disorders of JNK signaling. Unfortunately, this has not been done.”

A2. We would like sincerely to thank the reviewer for his/her comments. We will successively explain the above-mentioned issues as follows:

We have elucidated the regulating mechanism of JNK in this revision of our manuscript (paragraph 3, page 8). That is, The JNK signaling pathway can be disturbed by external factors such as cytokines, growth factor, ROS, pathogens, and drugs, etc. Abnormal expressions of some genes in JNK pathway or their mutations can also disturb JNK signaling. We simply summarized the activation or inactivation mechanisms for JNKs as follows: (1). The constitutively active mutants of MKK4/7 can cause JNK sustained activation while their dominant-negative mutants can inactivate JNK kinase. (2). DUPS overexpressed in cells can lead to inactivation of JNK. (3). JIP-mimics peptide may inactivate JNK by interfering the interaction of JNK and JIP or affecting the subcellular localization of JNK. For example, over-expressed JIP-1 can inhibit JNK MAPK signalling either by inhibiting JNK activity or by altering subcellular localization [Int J Biochem Cell Biol. 2001, 33(11):1047-63]. (4). The constitutively active mutations of some upstream kinases (e.g. TAK1) or other molecules ( e.g. agouti-related peptide) can activate JNKs (AgRP) [J Biol Chem. 1998, 273(10):5423-6; Cell Rep. 2014;9(4):1495-506]. (5). Small molecule inhibitor can inactive JNK kinase by occupating and inhibiting the kinase catalytic center or interfering the JNK-JIP interaction. We have provided the above information about regulating mechanism of JNK activation and revised the corresponding paragraphs in the text. The corresponding descriptions were highlighted by underlining and in red color in this revision of the manuscript (Paragraph 3, page 8).  Due to the complexity of the above-mentioned factors affecting JNK activity, we don’t think that they are tissue-specific expression.

It is now clear that any one cell, whether it be a simple yeast cell or a more complex mammalian cell, may contain MAPKs from several distinct subfamilies, indicating the wide expressions of JNK in cells [Int J Biochem Cell Biol. 2001 Nov;33(11):1047-63]. Although JNK MAPKs are located in both the cytoplasm and nucleus of quiescent cells, the activated JNK MAPKs can translocate to the nucleus [Int J Biochem Cell Biol. 2001 Nov;33(11):1047-63]. Specifically, JNK1 and JNK2 mainly occurs in cytoplasm and nucleus [Mol Biol Cell. 2011, 22(1):117-27; PLoS One. 2009 Aug 13; 4(8):e6640.]. JNK3 can occur in cytoplasm, nucleus, membarane and mitochondrion [J Biol Chem. 2006 Jul 28;281(30):21491-21499]. Based on these findings, we have provided the above information about the cellular localizations of JNKs and revised the corresponding paragraphs in the text. The corresponding descriptions were highlighted by underlining and in red color in this revision of the manuscript (Paragraph 2, page 4).

In this review, we classified JNK inhibitors as the following three  types of inhibitors: ATP-competitive inhibitors, ATP non-competitive inhibitors, and small peptide inhibitors. Mechanistically, ATP competitive inhibitors interact with the hinge region of the ATP binding site. The non-competitive inhibitors do not bind to ATP-binding sites but bind to the catalytic sites, indicating that the non-competitive of inhibitors inhibit JNK by inhibiting the kinase activity. By contrast, small peptide inhibitors regulate JNK kinase activity by interacting with  JNK’s scaffold or its upstream/downstream molecules or by changing the subcellular localization. In fact, we briefly summarized the classification of JNK inhibitors and their action mechanism in our previous manuscript (Paragraph 4, page 19). According to your comments and another reviewer’s (reviewer 2)  comments, we added  the corresponding action mechanisms of specific JNK inhibitors in section 4.2 (4.2 The application of synthetic JNK inhibitors in human diseases) in this revision of our manuscript. The corresponding revisions were highlighted by underlining and in red color (from page 20 to page 22).

As you said, JNKs play a role in almost all human diseases. Theoretically, a common strategy for treating thes diseases with the comman target JNK would be desirable. For example, JNK inhibitor SP600125 can be used to treat several diseases such as ischemia/reperfusion, cancer, inflammation, viral infections and sepsis-induced lung injury, etc. However, this would require the identification of common or individual disorders of JNK signaling. We think that identification of common or individual JNK signaling disorder under some disease states may be a future research direction. We think that it is reasonable to add this discussion to section 5 (5. Conclusions and perspectives). Therefore, we have added a paragraph to discuss this common strategy in section 5 in this revision of our manuscript. The corresponding revisions were highlighted by underlining  and in red color (paragraph 2, page 23).

Q3.“The manuscript also contains a number of formal errors, both in the illustrations and in the text. The English grammar needs to be revised by a native English speaker.

A3. We would like sincerely to thank the editor for his/her comment. According to the comment above, we have thoroughly proofread the manuscript with the help of a professor in our lab, and have corrected the corresponding grammar issues and typos. The corresponding revisions were highlighted by underlining  and in red color in this revision of the manuscript. 

Q4.“The following points are not a complete list of weaknesses, but serve only to illustrate some flaws.

Lines 47 and 53 are a duplication.”

A4. We would like sincerely to thank the reviewer for his/her comments. According to the comment above, we have deleted the sentence "In general JNK signaling is strictly regulated within cells" in L53 in this revision of our manuscript. The corresponding revisions were highlighted by underlining  and in red color in this revision of our manuscript  (paragraph 2, page 2).

Q5. “In Fig. 1 B, the information on 2004 must be reformulated. The time line is a continuum and can therefore not be A and B. In Fig. 1 B, the information on 2004 must be reformulated”

A5. We would like sincerely to thank the reviewer for his/her professional suggestion. According to the comment above, we have reformulated the information on 2004 in a revised figure 1 and deleted panel A and B, and replaced the previous figure 1 with this revised figure 1, and revised the corresponding figure 1 legend in the text. The corresponding revisions were highlighted by underlining  and in red color in this revision of our manuscript  (paragraph 1, page 3).

Q6. “Row 44 and rows 84- 85 are repetitions”

A6. We would like sincerely to thank the reviewer for his/her comment. According to the comment above, we have deleted the sentence "As mentioned above, JNK1 (MAPK8), JNK2 (MAPK9) and JNK3 (MAPK10) are encoded by Jnk1, Jnk2 and Jnk3 genes, respectively." (previous Rows 84-85) in this revision of our manuscript. The corresponding revisions were highlighted by underlining  and in red color in this revision of our manuscript  (paragraph 2, page 3).

Q7. “Line 102: Which kinase is meant? ”

A7. We would like sincerely to thank the reviewer for his/her comment. We are sorry for our unclear description about the word “kinase”.  The word “kinase” in line 102 means MAP2K such as MKK4 and MKK7. We have added the word “MAP2K” before the word “kinase” in the text (Line 144, section 2.3). The corresponding revisions were highlighted by underlining  and in red color in this revision of our manuscript (the last paragraph, page 4).

Q8. “Legend of Fig. 2 is incomplete. Information about the bound ligands and an indication in the figure is necessary.”

A8. We would like sincerely to thank the reviewer for his/her comment. According to the comment above, we have added the information of the bound ligands for JNK1 (panel A and B), JNK2 (panel C) and JNK3 (panel D),  and indicated the corresponding ligands in the revised figure 3 (previous figure 2). In addition, we have supplemented the corresponding description in Figure 2 legend. The corresponding revisions were highlighted in Figure 2 legend in this revision of our manuscript (paragraph 2, page 5).

Q9. “ Lines 117-119: What is known about the activation of JNKs by viral proteins?”

A9. We would like sincerely to thank the reviewer for his/her comment. We briefly summarized  that JNK pathway can be activated by some viruses such as influenza A virus and HIV-1, etc., in “3.5.1 JNK in viral diseases” section in our previous manuscript. Mechanistically, most of viruses use their encoding proteins to activate JNK signaling. For instances, the NS1 protein of influenza A virus can activate JNK. Moreover, the amino acid residue phenylalanine (F) at position 103 of NS1 is decisive for JNK activation [J Virol. 2014, 88(16):8843-52; Cell Microbiol. 2017, 19(7).  doi: 10.1111/cmi.12721]. Similarly, the Tat protein of HIV-1 (Human immunodeficiency virus, Type 1) can activate JNK signaling through a Nox2-dependent manner. [J Biol Chem, 2007. 282(52): p. 37412-9].  SARS-CoV-2 virus also uses its spike proteins to activate JNK pathway through toll-like receptor signaling [Cureus. 2022 Dec 9;14(12):e32361]. Based on these findings, we think that viral structural protein is critical for activation of JNKs though the mechanism  is not fully understood. According to the comment above, we have supplemented the above information about the activation of JNKs by viral proteins in section 2.4 (2.4 General activation mechanism of JNK signaling pathway) in the text (Previous lines 117-119). The corresponding revisions were highlighted by underlining  and in red color in this revision of our manuscript (paragraph 2, page 7).

Q10. “Fig. 3: The MAP3K, MAP2K and MAP3K line is not integrated in the Fig. Is the numbering correct?”

A10. We would like sincerely to thank the reviewer for his/her comment. We are sorry for our carelessness. The MAP3K, MAP2K and MAP3K line in the previous figure 3 should  be MAP3K, MAP2K and MAPK line, i.e,  JNK corresponds to MAPK. Thus we revised the word “MAP3K” in to “MAPK” in this revised figure 4.  In addition,  in regarding to  the integration of the MAP3K, MAP2K and MAP3K line, what we mean in the previous figure 3 is that JNKKKs, MKK4/7 and JNK1/2/3 corresponds to MAP3K, MAP2K and MAPK, respectively. In order to avoid misunderstanding about figure 4, we indicated them by using a dashed line. 

Q11. “Figure 3 suggests a pure signaling cascade. Are there other kinases or other cellular or viral proteins that phosphorylate or bind to members of the signaling cascade shown?”

A11. We would like sincerely to thank the reviewer for his/her professional suggestion. After carefully reviewing the literatures, we found that several STE20 protein homologue germinal center kinase family memebers including HPK, NIK and GCK can interact with and phosphorylate MEKK1. Additionally, several protein complexes such as ACID-JIP1-JNK1, JNK1/2/3-Sab and Notch1-IC-JIP1-JNK1/2/3 complexes are also involved in JNK signaling transduction, that is, ACID (amyloid precursor protein intracellular domain), Sab and Notch1-IC (Notch intracellular domain) can affect JNK signaling by binding to JIP1-JNK1 complex, JNK1/2/3, and JIP1-JNK1/2/3 complex, respectively. Thus, we have supplemented the corresponding diagrams of  these kinases or proteins in JNK signaling cascade  in this revised figure 4. We also revised the figure 4 legend and the corresponding paragraph in section 2.4 (section 2.4 General activation mechanism of JNK signaling pathway) in the text. The corresponding revisions were highlighted by underlining  and in red color in this revision of our manuscript (paragraph 1, page 6).

Q12. “ Lines 173-175: a citation is missing”

A12. We would like sincerely to thank the reviewer for his/her comment. Because the contents of  lines 173-175 (current lines 294-296) were cited from the literature (Cells. 2020 Mar 13;9(3):706.), we have inserted the reference (Ref. 58) in the references section in this revision of the manuscript.

Q13. “A protein phosphatase is mentioned in the text, but not included in the figure. Are there other phosphatases that interfere with the signaling cascade?”

A13. We would like sincerely to thank the reviewer for his/her professional comment. We mentioned that dual specificity protein phosphatase (DUSP1) deactivates JNK pathway through dephosphorylation of JNK in the text. Actually, JNK can be dephosphorylated by Ser phosphatases such as PP2A, and more specifically by dual specificity phosphatases (DUSPs) including DUSP1, DUSP4, DUSP10, DUSP16 and DUSP12 [Curr Med Sci. 2018 Feb;38(1):115-123; Cell Signal. 2021; 84:110033.], implying that these phosphatases can interfere with the signaling cascade. Thus we have supplemented the corresponding diagram of PP2A- and DUSPs-mediated dephosphorylation of JNK  in this revised figure 4. And we also added the corresponding description in both figure 4 legend and 2.4 section (General activation mechanism of JNK signaling pathway) in the text. The corresponding revisions were highlighted by underlining and in red color in this revision of our manuscript (the last paragraph, page 6).

Q14. “Figure 4 A-D is systematically wrong. Only IRS1/2 phosphorylation is shown, Akt phosphorylation is missing. What means “Insulin action? In B- D transcription or transcribed genes are shown. The text in lines 197 - 208 is only incompletely reflected in Fig. 4, if at all. ”

A14. We would like sincerely to thank the reviewer for his/her professional comment. We are sorry for our carelessness for missing Akt phosphorylation. “Insulin action” means the metabolic action of insulin. Therefore, we have supplemented the corresponding schematic diagram of Akt phosphorylation, and replaced “Insulin actin” with “Insulin metabolic actin” in this revised figure 5. In addition, we mainly discussed the effects of IRS1 phosphorylation by JNK on PI3K-AKT signaling and insulin resistance in the text in previous lines 197- 208 (current Lines 218-329; the last paragraph, page 9). In the next two paragraphs, we discussed the other three molecular mechanisms for the involvement of JNK pathway in insulin resistance (Fig.5 B-D). However, we did not provide a detailed description about the transcription process or transcribed genes. According to the comment above, we have added the corresponding descriptions about the transcription process or transcribed genes in Figure 5 B, C and D. The corresponding revisions were highlighted by underlining and in red color in this revision of our manuscript (the last paragraph, page 10; paragraph 2, page 11).

Q15. “Lines 268- 269: It is not clear which function of JNK plays an important role”

A15. We would like sincerely to thank the reviewer for his/her professional comment. Atherosclerosis onset and progression involves the following events: (1). An initiating stimulus including vascular injury, hypercholesteremia, or chronic inflammation. These stimuli cause endothelial cell dysfunction and/or apoptosis, thus increasing vessel wall permeability to lipids and localized inflammation. Under these conditions, endothelial cells express cytokines that recruit monocytes and leukocytes to the area; (2). Monocytes then transmigrate across the blood vessel wall and differentiate into macrophages. As the vessel is lipid-laden, the macrophages ingest the low-density lipoproteins (LDLs) then turn into foam cells. These foam cells further promote lesion progression by increasing local inflammation and ROS production; (3). Under the action of endothelial cell-secreted cytokines, T and B lymphocytes are recruited to the plaque. Smooth muscle cells are also recruited to the lumen and begin to proliferate and secrete collagen, elastin, and other extracellular matrix proteins. While multiple cell types contribute to this complex process, elevated JNK activity was observed in each of these steps. It is well known that JNK1/2 plays an important role in inflammation and cytokine production. Moreover, JNK1/2 is expressed in all of the cell types relevant to the onset and progression of atherosclerosis: endothelial cells, smooth muscle cells, macrophages, and T cells, all of which are associated with the formation and development of atherosclerosis. In addition, previous studies have shown that JNK1 promotes apoptosis in endothelium after chronic inflammation and promotes atherosclerosis. Similar to endothelium, JNK1 in bone marrow-derived immune cells (including monocytes) also promotes apoptosis after chronic inflammation which leads to less atherosclerosis in mice. JNK2 knock-out mice showed less atherosclerosis through reduced number of foam cell formation. Babaev et al. demonstrated that hematopoietic cell-specific JNK1 deficiency promoted atherosclerosis in LDLR−/− mice. By contrast, Nofer et. al. found that JNK promotes atherosclerosis onset [Biosci Rep. 2019 Jul 18;39(7):BSR20190267; Arterioscler Thromb Vasc Biol 27(8) (2007) 1677-8; Science 306(5701) (2004) 1558-61; Atherosclerosis 235(2) (2014) 613-8]. These findings suggest that JNK plays an important role in inflammation and cytokine production and mediates the apoptosis of endothelium, immune cells and foam cells during the onset and progression of atherosclerosis, and that JNK has different functions in different cells (macrophages, T-cells, endothelial cells etc.). In the text, we have added the descritions about the above contents to explan the specific function of JNK during the onset and progression of atherosclerosis in section 3.1.3. The corresponding revisions were highlighted by underlining and in red color in this revision of our manuscript (the last paragraph, page 11).

Q16. “Line 293: What is meant by "over-activation of JNK pathway..."?  Is the kinase activity activated or is there an elevated level of JNKs, or both?

A16. We would like sincerely to thank the reviewer for his/her comment. The sentence "over-activation of JNK pathway..." in previous line 293 (current line 441) means that the kinase activity of JNK is activated in CNS pathologies such as glioblastoma and Alzheimer disease, etc. For instances, previous studies have shown that the JNK kianse is activated in Glioblastoma (GB) cells and promotes GB progression and infiltration [PLoS Biol. 2019 Dec 17;17(12):e3000545; Cell Death Dis. 2018 Mar 12;9(3):394]. Moreover, Portela et. al. used an Tre-RFP reporter, which undergoes transcriptional activation in response to JNK signaling, to show an up-regulation of Tre-RFP reporter, thus verifying that JNK is activated in GB cells. They found that the increase in JNK activity depends on the presence of Fz1 receptor in GB cells. In addition,  they also found that GB cells establish a positive feedback loop to promote their expansion, in which the Wingless (Wg) pathway activates cJun N-terminal kinase (JNK) in GB cells, and, in turn, the activated JNK leads to the post-transcriptional up-regulation of matrix metalloproteinases (MMPs), which facilitate GB infiltration throughout the brain  [PLoS Biol. 2019 Dec 17;17(12):e3000545]. Li et al. collected different glioma samples from patients exposed to surgery to study the correlation of the JNK’s activity with the tumor progression, and found that constitutive activation of JNK correlates with histologic grade in diffuse gliomas [J. Neurooncol. 2008, 88, 11–17]. Similarly, Sclip et al. found that JNK is activated at the spine prior to the onset of cognitive impairment, indicating JNK has a key role in Alzheimer disease synaptic dysfunction in vivo [Cell Death Dis. 2014, 5, e1019]. Based on these findings, we think that the sentence “over-activation of JNK pathway…” means the increase of JNK kinase activity. We remarked the meaning of the sentence in this revision of our manuscript. The corresponding revisions were highlighted by underlining and in red color in this revision of our manuscript  (paragraph 3, page 12).

Q17. “ What means  “ The JNK signaling in CNS… is stronger”?”

A17. We would like sincerely to thank the reviewer for his/her comment. We apologize for not expressing our thoughts clearly. What we mean is that the JNK pathway is highly active in the CNS as compared with other tissues [Int J Mol Sci. 2021 Apr 9;22(8):3883]. Thus we revised the sentence in section 3.2.1  (Line 447) in this revision of our manuscript. The corresponding revisions were highlighted by underlining and in red color in this revision of our manuscript  (paragraph 4, page 12).

Q18. “Lines 324 “JNK also mediates PD progression..” Does JNK this alone or together with other factors? What is the molecular basis for this mediation?”

A18. We would like sincerely to thank the reviewer for his/her comment. After carefully reading the literatures, we found that the mechanism by which JNK mediates PD progression can be summarized as follows: JNK-mediated autophagy and apoptosis play key roles in PD progression. During this process, Bcl-2 was identified as a critical protein with the ability to suppress autophagy and apoptosis through inhibiting Beclin-1 and Bax, respectively. In terms of apoptosis, both JNK and P38 mediate dopaminergic neuron apoptosis in PD by increasing the Bax/Bcl-2 ratio. In regarding to autophagy, JNK-mediated BCL-2 phosphorylation also suppresses the functions of Bcl-2 in autophagy. Additionally, recent findings indicate that the receptor-interacting protein kinase 1 (RIPK1) is upregulated in PD in vitro and in vivo models. Importantly, RIPK1 promotes cell apoptosis and reactive oxygen species (ROS) production through the activation of the JNK pathway, thus leading to dopaminergic cell death. These discoveries suggest that JNK together with other factors such as p38, BCL-2, Bax, Beclin-1, and RIPK1 mediate PD progression [Neurobiol Aging, 2021, 100: 91-105; J Neuropathol Exp Neurol, 2010, 69(5): 511-20]. Therefore, we have added the related descriptions about the above mechanism into section 3.2.2 (3.2.2 JNK pathway in neurodegenerative diseases) in the text. The corresponding revisions were highlighted by underlining and in red color in this revision of our manuscript  (paragraph 2, page 13).

Q19. “Lines 345- 350 there is paper cited which showed an upregulation of phosphorylated JNK. This is an indication that not only JNK functions as a kinase but it is itself phosphorylated by one or more protein kinases. The kinases which phosphorylate JNKs would give an important inside view into an upstream regulation of JNKs. ”

A19. We would like sincerely to thank the reviewer for his/her comment. After carefully reading the literatures, we found that during MS progress, the phosphorylation of JNK by ASK1 plays critical roles in mutile processes including oligodendrocyte destruction, demyelination, neuroinflammation, immune dysregulation through T cell activation (T cell apoptosis) and oxidative damage. For instances, the TLR-ASK1-JNK pathway is active in glial cells and important for autoimmune demyelinating disorders [EMBO Mol Med 2(12):504–515]. Besides TLR (Toll-like receptor), other factors including reactive oxygen species (ROS), oxidative stress, and inflammation that can activate ASK1 in the pathogenesis of MS [Neurotox Res. 2021 Oct;39(5):1630-1650; Antioxid Redox Signal 2002, 4:415–425; Int J Mol Sci. 2021 Apr 9;22(8):3883]. These reports indicated that ASK1-MKK4/7-JNK signaling axis plays critical role in the pathogenesis of MS. Thus, the kinases which phosphorylate JNKs is the JNK upstream MAPK kinases (MKK4 and MKK7). However, the upstream MAPK kinase of MKK4/7 kinase is apoptosis signal-regulating kinase 1 (ASK1), which is very important for MS progression. In the text, we have briefly summarized the critical roles of ASK1-MKK4/7-JNK axis in pathogenesis of MS. The corresponding revisions were highlighted by underlining and in red color in this revision of our manuscript  (the last paragraph, page 13; paragraph 1, page 14).

Q20. “Figure 5 would be a good summary but the authors must explain what activation and inactivation means.”

A20. We would like sincerely to thank the reviewer for his/her comment. In figure 5 (current figure 6), activation represents sustained activation of JNK pathway while inactivation means JNKs deficiency or inhibition of JNK by JNK inhibitor such as SP600125. According to the coment above, we added the descriptions about the meanings of activation and inactivation in the figure 6 legend, and the corresponding revisions were highlighted by underlining and in red color in this revision of our manuscript (paragraph 1, page 16).

Q21. “Line 605: “ .. as mentioned above, different diseases involve different JNK isoforms… “. This is a very interesting point which is not consequently addressed in the review.”

A21. We would like sincerely to thank the reviewer for his/her comment. We mentioned that different diseases involve different JNK isoforms (previous line 605). It is difficult for us to cover all aspects of JNK's progress in such an article, thus we just took JNK3 as a example to illustrate our opinion. We have added a sentence “JNK3 is mainly involved in CNS-related diseases because of its brain-specificity” in the text. The corresponding revision was highlighted in this revision of our manuscript (Line 790-791 in section 4.2.1, paragraph 1, page 21).

Q22. “With regard to inhibitors, information about the specificity, isoform specificity, off target effects etc. would be very helpful. Are there any crystal structures known for JNKs alone or the complex with inhibitors?”

A22. We would like sincerely to thank the reviewer for his/her comment. We  have shown the structure of JNK1 (Fig 2C, Fig 3A and B), the structure of JNK2 (Fig 3C) and the structure of JNK3 (Fig 3D) in figure 4 in this review. According to the comment above, in this revision, we added a new paragraph (4.4. The challenges we faced when we are developing JNK inhibitors) to explan the significance of  the information about the specificity, isoform specificity and off target effects of JNK inhibitors (paragraph 4, page 22), and summarized the (isoform) specificity and off-target effects of several JNK inhibitors in a new table (table 2). The structural information of JNK complexed with JNK inhibitors (PDB ID) were also provided in this new table 2. Table 2 was inserted in section 4.4 in this revision of our manuscript (from page 22 to page 23).

Round 2

Reviewer 3 Report

Comments and Suggestions for Authors

Second review on the manuscript entitled „The role of dysregulated JNK signaling pathway in the pathogenesis of human diseases and its potential therapeutic strategies: a comprehensive review“ by Yan et al.

The revised version of the manuscript is significantly improved compared to the first version. The reviewers' points of criticism have essentially been taken into account. It is still not always clear whether the activation of JNKs is due to an increased kinase activity or due to an elevated level of JNK proteins.

The manuscript still has weaknesses in the English language. This applies in particular to the changes made in red.

The word "microphage" is used in Fig. 5. The authors need to explain what they mean by this. Do they mean macrophages? The capitalization in the figures needs to be checked (e.g. Fatty Acid, Microphage in Adipose tissue).

The use of "et al." is inconsistent. Sometimes et. al. (line 474 and others) is used, sometimes et al

Line 484 must read: …. in in vitro and in in vivo ……..

Line 522 must read: „this tissue“

Line 777 must read: inhibitor instead of inhibitors

The following references are not complete. As a rule, the page numbers are missing.

[28], [30], [33], [36], [47], [49], [58], [91], [100], [104,],  [106], [111], [112], [113], [125], [136]

Comments on the Quality of English Language

Moderate editing of English language is required

Author Response

Dear editor,

On behalf of the authors I would like sincerely to thank you for your reading our manuscript and give our appreciation to the reviewers for their comments on our work.

Enclosed is our revised manuscript entitled “The role of dysregulated JNK signaling pathway in the pathogenesis of human diseases and its potential therapeutic strategies: a comprehensive review. Our previous manuscript No.is biomolecules-2785427. Based on the reviewer’s comments we revised our manuscript as follows:

Reviewer’s comments: Reviewer 3

Q1. “The revised version of the manuscript is significantly improved compared to the first version. The reviewers' points of criticism have essentially been taken into account. It is still not always clear whether the activation of JNKs is due to an increased kinase activity or due to an elevated level of JNK proteins.”

A1. We would like sincerely to thank the reviewer for his/her comment. To answer this question, we have carefully reviewed many references, and draw a conclusion that the activation of JNKs is due to an increased kinase activity. However, the expression level of JNK may has an effect on the activation of JNKs. Our opinion is supported by the following evidences:

(1). Previous studies have shown that the catalytic activity of MAPK protein kinases is tightly regulated through the activation segment that may be phosphorylated by other kinases to facilitate catalytic activity (Mol Cell, 2004, 15(5): 661-675; J Biol Chem, 2012; 287(8):5446-58). In general, a kinase will phosphorylate the activation segment of a downstream kinase, allowing the downstream kinase to further propagate a signal. Importantly,  phosphorylation of the activation segment at the primary phosphorylation site stabilizes the kinase in a conformation suitable for substrate binding. Kinases may also have secondary phosphorylation sites in their activation segment, which may enhance its activity. Secondary phosphorylation may also aid in the recruitment of substrate by changing the conformation of a kinase to facilitate substrate binding. Once a kinase is stabilized and activated, the catalytic domain identifies a specific substrate and phosphorylates the substrate via its active site  (Mol Cell, 2004, 15(5): 661-675). In other words, phosphorylation of the activation segment plays an important role in the regulation of kinase activity. For example, the Ste20-like kinase (SLK), which is a JNK upstream kinase, can phosphorylate JNK kinase. Luhovy et al. found that, compared with wild-type SLK (SLK-WT), mutations in serine (Ser189) and threonine (Thr183) residues in the activation segment of SLK significantly reduced its kinase activity. That is, the T183A, S189A, and T183A/S189A mutants showed a reduced in vitro kinase activity of SLK. SLK-WT, but not mutants, increased activation specific phosphorylation of JNK kinase (J Biol Chem, 2012; 287(8):5446-58). These findings suggest that the phosphorylation of serine (Ser189) and threonine (Thr183) residues in the activation segment of SLK determines its kinase activity, which is responsible for phosphorylating  its downstream JNK.

(2).  As we summarized in our manuscript, JNK belongs to the mitogen-activated protein kinase (MAPK) family. JNK activation requires the phosphorylation of both Thr and Tyr residues in its Thr-Pro-Tyr motif in the activation loop between VII and VIII of the kinase domain. The activation loop is similar to the activation segment of SLK. For example, JNK1 is activated when dual phosphorylation occurs on Thr-183 and Tyr-185 in the Thr-Pro-Tyr motif. The phosphorylation is catalyzed by the dual specificity kinases MKK4 and MKK7. Furthermore, the Tyr-185 and Thr-183 residues of JNK are sequentially phosphorylated by MKK4 and MKK7, respectively. Interestingly, MKK4 shows a striking preference for the tyrosine residue (Tyr-185), and MKK7 shows a significant preference for the threonine residue (Thr-183) (J Biol Chem, 2003, 278(19):16595-601; Biochem J, 2000, 352 Pt 1(Pt 1):145-54; Redox Biol. 2014:3:7-15). From these findings, we can draw a conclusion that similar to SLK, activation of JNKs also requires the phosphorylation of both Thr and Tyr residues in the activation loop. In fact, we have mentioned this key point in  section 2.4 in  previous version of our manuscript (highlighted by underlining and in blue color in the last paragraph, page 5). In this second revision, we have provided a detailed description of the activation of  JNKs.  

(3). Previous studies have shown that the phosphorylation of JNK at Thr-183 and Tyr-185 (p-Thr183/Tyr185) represents its activation form, which exerts the kinase activity of JNK and can phosphorylate its substrates including c-Jun (J Affect Disord. 2020 Nov 1:276:626-635; Cell Death Dis. 2018 Apr 1;9(4):421. doi: 10.1038/s41419-018-0459-3.; Nat Commun. 2022, 13(1):6133). For example, Christopher et al. have recently reported that the phosphorylated JNK shows the kinase activity on c-Jun, that is, the phosphorylated JNK (p-Thr183/Tyr185)  can phosphorylates the c-Jun protein at four residues within its transactivation domain (TAD). Among these residues, Ser-63 and Ser-73 are phosphorylated by JNK more rapidly than Thr-91 and Thr-93. They further reveals that different c-Jun phosphorylation states exert different functions: unphosphorylated c-Jun recruits the MBD3 repressor, Ser-63/73 doubly-phosphorylated c-Jun binds to the TCF4 co-activator, whereas the fully phosphorylated form disfavours TCF4 binding attenuating JNK signaling. Thus, c-Jun phosphorylation encodes multiple functional states that drive a complex signalling response from a single JNK input (Nat Commun. 2022, 13(1):6133).

(4).To our knowledge, no report has shown that the activation of JNKs is directly related to an elevated level of JNK proteins. However, Bildik et al. found that siRNA targeting JNK led to a significant downregulation of p-c-Jun (p-Ser63/Ser73). Importantly, siRNA targeting JNK also significantly downregulated JNK expression (Cell Death Dis. 2018 Apr 1;9(4):421. doi: 10.1038/s41419-018-0459-3), implying that the expression level of JNK (JNK protein amount) may has an effect on the activation of JNK. Additionally, Yue et al. found that JNK was activated during stress-induced apoptosis (Biochem Biophys Res Commun. 2003, 306(4):837-42; J Biol Chem. 2015, 290(51):30375-89), and sustained JNK activation accelerated the apoptotic program. Importantly, when caspase-3 was activated, JNK was proteolyzed at Asp-385 increasing the release of cytochrome c and caspase-3 activity, thereby creating a positive feedback loop (J Biol Chem. 2015, 290(51):30375-89). These studies also indicated that the amount of JNK protein might affect its activity.

Based on these findings, we think that the activation of JNKs is due to an increased kinase activity, however, an elevated level of JNK proteins may has an effect on the activation of JNKs. According to the comment above, we have supplemented a detailed information about the activation of JNKs, which is caued by an increased kinase activity, in the text. The corresponding descriptions were highlighted by underlining and in blue color in this second revision of the manuscript (from the last paragraph on page 5 to paragraph 1 on page 7). 

Q2. “The manuscript still has weaknesses in the English language. This applies in particular to the changes made in red.”

A2. We would like sincerely to thank the editor for his/her comment. According to the comment above, we have thoroughly proofread the manuscript, and have corrected the corresponding grammar issues and typos, especially those errors highlighted in red. The corresponding revisions were highlighted by underlining  and in blue color in this revision of our manuscript.

 Q3. “The word "microphage" is used in Fig. 5. The authors need to explain what they mean by this. Do they mean macrophages? The capitalization in the figures needs to be checked (e.g. Fatty Acid, Microphage in Adipose tissue).”

A3. We would like sincerely to thank the reviewer for his/her comment. We apologize for our carelessness. The word "microphage" that is used in Fig. 5 means macrophages. Thus we have corrected it to  "macrophage" in this revised figure 5. In addition, we have carefully checked the capitalization in the figure 5, and revised “Insulin Receptor” in panel A and “Fatty Acid oxidation” in panel D to “Insulin receptor” and “Fatty acid oxidation”, respectively, and have replaced previous figure 5 with a corrected figure 5.

We have also carefully checked the capitalization in othe figures, and found that “JNK2 Crystal structure……(2008-2009)” in figure 1  should be “JNK2 crystal structure”,  and that “Multiple Myeloma” in panel A of figure 6 should be “Multiple myeloma”. There fore, we have corrected these capitalizations, and replaced previous figure 1 and figure 6 with  two corrected figure 1 and figure 6.

 Q4. “The use of "et al." is inconsistent. Sometimes et. al. (line 474 and others) is used, sometimes et al.”

A4. We would like sincerely to thank the reviewer for his/her comment. In this revision, we have used the consistent "et al.". The corresponding descriptions were highlighted by underlining and in blue color in this revision of the manuscript.

Q5. “Line 484 must read: …. in in vitro and in in vivo ……..”

A5. We would like sincerely to thank the reviewer for his/her comment. According to the comment above, we have revised “in vitro and in vivo”  in previous line 484 to “in in vitro and in in vivo” in line 544. The corresponding descriptions were highlighted by underlining and in blue color in this revision of the manuscript (paragraph 2, page 14).

 Q6. “Line 522 must read: this tissue”

A6. We would like sincerely to thank the reviewer for his/her comment. According to the comment above, we have revised “these tissue”  in previous line 522 to “this tissue” in line 582. The corresponding revision was highlighted by underlining and in blue color in section 3.2.5 in this revision of the manuscript (paragraph 2, page 15).

 Q7. “Line 777 must read: inhibitor instead of inhibitors”

A7. We would like sincerely to thank the reviewer for his/her comment. According to the comment above, we have revised “inhibitors”  in previous line 777 to “inhibitor” in line 837. The corresponding revision was highlighted by underlining and in blue color in section 4.2.1 in this revision of the manuscript (paragraph 1, page 22).

 Q8. “The following references are not complete. As a rule, the page numbers are missing.

[28], [30], [33], [36], [47], [49], [58], [91], [100], [104],  [106], [111], [112], [113], [125], [136].”

A8. We would like sincerely to thank the reviewer for his/her comment. According to the comment above, we have added the page numbers of the above references. The corresponding revisions were highlighted by underlining and in blue color in the references section in this revision of the manuscript.

The revised version was upload to the submitting system. We hope that you will be satisfied with this second revised version of the manuscript. Please feel free to mail to me if you have any question.

Thank you very much for reading our revision of the manuscript again. I am looking forward to hearing from you soon.

Sincerely yours,

Zhu Yuan

State Key Laboratory of Biotherapy

West China Hospital

Sichuan University

Chengdu, Sichuan 610041, China;

Phone: (86) 28-8516-4063

E-mail: yuanzhu@scu.edu.cn